# Characterising Overprecision in Black-Box LLMs: A Cognitive Science Inspired Framework

## Abstract

Overconfidence in large language models (LLMs) has attracted growing attention due to its implications for the reliability of model outputs. Most existing approaches study verbalized confidence, where LLMs are asked to state their certainty, but such methods are prone to biases and hallucinations. Inspired by the cognitive science notion of overprecision, excessive certainty in narrow interval judgments, we propose a framework for evaluating overprecision in black-box LLMs. Our protocol comprises three phases: (1) generation, where models produce numerical confidence intervals under imposed confidence levels; (2) refinement, where intervals are adjusted via aggregation or self-refinement strategies; and (3) evaluation, where outcomes are assessed using calibration and correlation metrics adapted from cognitive science. Using datasets spanning general knowledge, medical, and financial domains, we find that: (i) LLMs are systematically miscalibrated, with large gaps between imposed confidence and actual coverage; (ii) interval lengths do not scale with requested confidence, showing limited responsiveness to explicit confidence instructions; (iii) calibration quality varies by task, domain, and answer scale, with finance and medicine posing greater challenges than general knowledge; and (iv) refinement helps only when it trivially widens (union); reflective self-refinement tends to narrow and can worsen coverage. Taken together, these findings show that miscalibration persists across settings. This work is an exploratory, descriptive study: our goal is to characterize black-box LLM behavior under a fixed protocol, not to benchmark or optimize models for maximal performance.

## 1 Introduction

Overconfidence is a cognitive bias that affects human decision-making, characterized by a level of confidence that exceeds what is justified by reality. In cognitive science, overconfidence has been studied across three distinct dimensions (Moore & Dev, 2017; Moore & Schatz, 2017): (1) Overestimation, (2) Overplacement, and (3) Overprecision. Overestimation involves an inflated perception of one's abilities or performance relative to their actual level. Overplacement refers to an exaggerated belief in one's superiority over others. Overprecision is defined as unwarranted certainty in the accuracy of one's knowledge or beliefs. Among these dimensions, overprecision is considered the most robust (Moore et al., 2015b;a), as it consistently lacks contradictory findings across different studies, unlike the other aspects.

Recent work has begun examining overconfidence in large language models (LLMs) (Xiong et al.; Geng et al., 2024), especially due to risks in safety-critical settings such as medicine and finance. However, this research focuses mainly on overestimation by eliciting verbalized confidence and largely overlooks overprecision, despite its importance and complementarity.The widespread reliance on verbalized confidence for LLM calibration is undermined by two critical empirical issues. First, confidence expressions suffer from instability and prompt sensitivity, leading to inferior calibration performance because the stated certainty changes easily with minor phrasing changes (Geng et al., 2024; Xu et al., 2024). Second, LLMs are often inherently miscalibrated, meaning their verbalized scores provide inaccurate estimates that fail to reflect the true empirical frequencies of their correctness (Lyu et al., 2025). Studying overprecision in LLMs offers several advantages: (1) LLMs are instruction-followers, making explicit confidence enforcement in the prompt more aligned with their operational mode; (2) linguistic cognitive biases (e.g., positivity, order) (Sumita et al., 2024) distort verbal confidence more than they do numerical intervals; (3) overprecision tests understanding of

uncertainty principles, where higher confidence should yield wider intervals; and (4) interval-based evaluation better fits numerical reasoning tasks, which need not rely on exact correctness.

To address this gap, we propose a three-phase framework for studying LLM overprecision: **generation** of confidence-conditioned numerical intervals, **refinement** using cognitive- and LLM-inspired prompting techniques, and **evaluation** with cognitive-science-based calibration metrics (Figure 1). We contribute by: (1) introducing a protocol for eliciting interval-based confidence judgments, adapting cognitive science methods to numerical reasoning in LLMs; (2) applying this protocol across multiple datasets and prompting strategies to systematically evaluate overprecision without access to model internals; and (3) analyzing refinement strategies, showing that only union-based aggregation consistently improves coverage while self-refinement often fails. While grounded in cognitive science, our novelty lies in adapting interval-based overprecision measurement to black-box LLMs, offering new insights into how these models represent and externalize uncertainty. Throughout, we take a behavioral-measurement stance. Results are reported descriptively to illuminate patterns in coverage and interval responsiveness; they are not intended as competitive benchmarks or claims of model superiority.

Our results reveal that (i) LLMs are systematically miscalibrated, with large gaps between imposed confidence and actual coverage; (ii) interval lengths do not scale with requested confidence, showing limited responsiveness to explicit confidence instructions; (iii) calibration performance varies by task, domain, and answer scale, with finance and medicine posing greater challenges than general knowledge; and (iv) Refinement helps only when it trivially widens (union); reflective self-refinement tends to narrow and can worsen coverage. Taken together, these findings suggest that overestimation and overprecision represent distinct forms of overconfidence, implying that techniques successful in addressing one may not generalise to the other.

## 2  RELATED WORK

### 2.1  OVERCONFIDENCE IN HUMANS

Overconfidence—an unwarranted certainty in one's knowledge or abilities (Kruger & Dunning, 1999)—has negative consequences across domains such as medicine (Al-Maghrabi et al., 2024; Seidel-Fischer et al., 2024), politics (Ortoleva & Snowberg, 2015), and finance (Grežo, 2021). It is typically studied along three dimensions: overestimation, or inflated self-assessed ability measured through item-confidence judgments (Harvey, 1997); overplacement, the "better-than-average" effect where most rate themselves above others (Beer & Hughes, 2010); and overprecision, excessive certainty in one's estimates, assessed by asking participants to provide confidence intervals (Alpert & Raiffa, 1982). Of these, overprecision is the most robustly demonstrated across studies, while overestimation and overplacement often yield inconsistent results (Moore et al., 2015b;a). Our work therefore, focuses on overprecision, adapting interval-based methods from cognitive science to evaluate this phenomenon in black-box LLMs.

### 2.2  OVERCONFIDENCE IN LLMS

In cognitive science, overprecision is most commonly studied through interval judgments on numerical questions, making numerical reasoning tasks a natural setting for adapting these ideas to LLMs. Overconfidence in LLMs has been studied extensively (Geng et al., 2024). Existing approaches can be broadly grouped into two categories: white-box and black-box. White-box methods leverage access to internal model signals—such as logits or uncertainty estimates—to assess calibration and confidence (Huang et al., 2024; Duan et al.). By contrast, black-box methods operate without internal access, instead relying on prompting strategies to elicit self-reported confidence (Manakul et al., 2023; Mielke et al., 2022; Xiong et al.) or on surrogate models to approximate uncertainty (Shrivastava et al., 2023). Our work falls into the black-box paradigm.

#### 2.2.1  OVERCONFIDENCE IN BLACK-BOX LLMS

Most prior black-box studies of LLM overconfidence focus on the overestimation dimension, asking models to verbalize how certain they are about their answers (Wen et al.; Xiong et al.; Geng et al., 2024). While informative, this strategy has clear limitations. LLMs are optimized primarily for

instruction-following, not for explicitly modeling or reporting their internal uncertainty, and their training data only indirectly expose them to expressions of confidence. As a result, self-reported confidence can vary with prompt wording, sampling randomness, or linguistic bias, and may not reliably reflect the model's true uncertainty (Lin et al.). To mitigate these issues, we shift the burden of confidence specification from the model to the prompt, imposing explicit confidence levels and evaluating whether the resulting intervals align with them. This design leverages instruction-following strengths while avoiding the pitfalls of subjective self-assessment. We further restrict analysis to numerical answers, which enable finer-grained evaluation and reduce artifacts common in multiple-choice settings (**?**). Closest to our work, (Groot & Valdenegro-Toro, 2024) examined regression-style prompts for vision–language models; in contrast, we study interval-based overprecision in knowledge tasks, emphasizing calibration in numerical reasoning.

In summary, white-box calibration methods require internal signals (e.g., logits, variance proxies) and are inapplicable to closed models. Black-box methods that elicit verbal confidence suffer from linguistic and prompt-sensitivity artifacts (Lin et al.). We instead adapt interval-based overprecision measurement from cognitive science to the black-box LLM setting, testing coverage under imposed confidence levels and analyzing whether interval length responds to requested confidence.

## 3 OVERPRECISION IN BLACK BOX LLMS

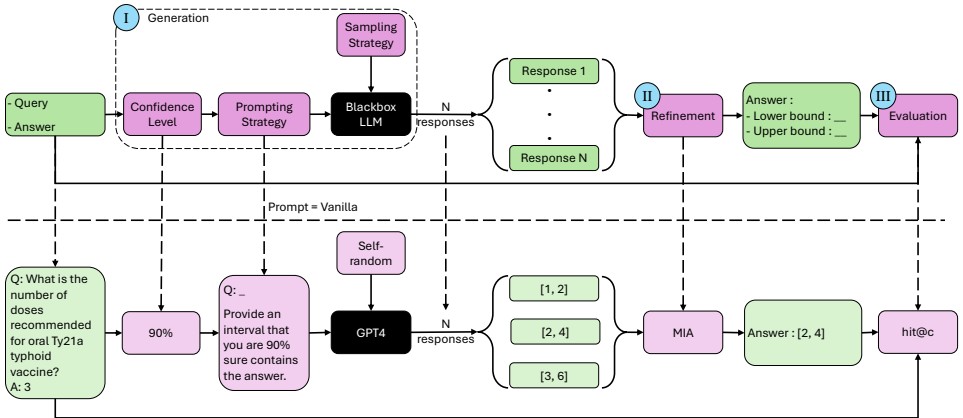

Figure 1: An outline of the precision elicitation framework and an example. Given an input question, a confidence level is first specified, a prompt strategy is then chosen, and the confidence level is integrated into the prompt. Next, the sampling strategy and the number of samples are determined to control the amount and diversity of outputs of the same prompt. After that, an *aggregator* combines the different answers to produce the most likely answer.

Let $\{(q_i, a_i)\}_i$ represent a set of questions and their corresponding answers, where $q_i$ is a textual question, and $a_i \in \mathbb{R}$ is its numerical answer. This work proposes a framework for studying overprecision in LLMs, consisting of three phases: (a) generation, (b) refinement, and (c) evaluation. The generation phase involves generating (i.e., predicting) an answer for each question using an existing LLM. The refinement phase takes the answers produced during the generation phase and applies various techniques to rectify and improve these answers. Finally, the evaluation phase analyzes the answers from the previous phases to assess the calibration and confidence responsiveness of the LLM. The details of each phase and its corresponding steps are presented in the following sections.

### 3.1 GENERATION

The objective of the generation step is to produce answers using an LLM. The generation process consists of two main components: (a) *prompting strategy* and (b) *sampling strategy*. The prompting strategy involves integrating the question into a confidence-parametrized prompt composed of various parts. This prompt, or its variants, is then provided to the LLM multiple times, following a specific sampling strategy. Formally, this phase is responsible for constructing a prompt $\mathbf{p}_c(q)$ parameterized

by a confidence level $c$. This prompt is fed into the LLM to generate a lower bound $x$ and an upper bound $y$, defining the interval within which the answer to the question $q$ should fall:

$$(x, y) = \text{LLM}(\mathbf{p}_c(q)) \tag{1}$$

- **Prompting Strategy**: We designed two prompting strategies: vanilla prompts, which directly ask the model to provide an interval at a specified confidence level, and chain-of-thought (CoT) prompts, which additionally request step-by-step reasoning before producing the interval. Both strategies explicitly embed the desired confidence level in the prompt. Full instruction templates are provided in Appendix B.
- **Sampling Strategy**: we sample via a self-random setting, where the same prompt is issued multiple times to capture stochastic variation in outputs.

## 3.2 REFINEMENT

To assess whether post-processing can mitigate overprecision, we evaluated two refinement strategies: aggregation and self-refinement.

- Aggregation. This strategy combines multiple model-generated intervals into a single one using simple schemes—mean, length-weighted, inverse-length-weighted, and union. These methods test whether pooling diverse estimates improves coverage compared to individual predictions:
  - *Mean interval aggregation* (MIA) the lower and upper bounds across intervals.
  - *Length-weighted aggregation* (LWA) and *inverse-length-weighted aggregation* (iLWA) give more weight to longer or shorter intervals, respectively.
  - *Union aggregation* (Union) takes the minimal lower bound and maximal upper bound, maximizing coverage at the cost of informativeness.
  - *Confidence-weighted aggregation* (CWA) assigns greater weight to intervals generated at higher confidence levels, producing bounds that emphasize stronger certainty.
- Self-refinement. In this approach, the model is shown several of its previously generated intervals and asked to (i) select the most plausible one and (ii) optionally propose a new interval. This design is inspired by cognitive science findings that exposure to peer judgments can reduce overprecision (Haran et al., 2010), with the model here acting as its own "peer reviewer."

Aggregation methods are common in categorical prediction tasks but are rarely applied to interval outputs. We adapt them to test whether simple pooling rules can improve calibration. Self-refinement extends this idea by encouraging the model to reflect on its own predictions. While our design represents only one possible formulation, it provides a first step toward studying whether reflective prompting can mitigate overprecision[1]. We emphasize that both refinement strategies should be viewed as exploratory baselines rather than definitive solutions. Full mathematical definitions, weighting schemes, and prompt templates are provided in Appendix C.

## 3.3 EVALUATION

We evaluate calibration using the standard hit@c metric, which measures the proportion of cases where the true answer falls within an interval that the model claims should contain it with probability c (e.g., a 90% interval should succeed about 90% of the time under perfect calibration). We also compute the Pearson correlation between interval length and imposed confidence to test whether models widen intervals when asked to be less confident. Both measures follow established practice in cognitive psychology studies of overprecision (Moore et al., 2015a; Haran et al., 2010; Moore et al., 2015b). To provide a more nuanced view, we introduce two additional metrics: the Deviation Score (DS), which quantifies how far a missed answer lies outside the predicted interval, and the Interval Length Score (ILS), which normalizes interval width by the scale of the predicted interval bounds to penalize unnecessarily wide intervals. Formal definitions are given in Appendix D.

---

[1]We use calibration and coverage to describe how closely empirical hit rates match stated confidence levels. We use overprecision to denote the bias where intervals are too narrow for the claimed confidence, distinct from the statistical meaning of "precision" as inverse variance.

| dataset | #examples | avg-a | min-a | max-a |
|---------|-----------|-------|-------|-------|
| FinQA | 3262 | 1.109e+08 | -2.094e+09 | 8.096e+10 |
| Medical | 2058 | 4.033e+03 | -1.000e+02 | 6.123e+06 |
| MMLU | 1606 | 1.222e+10 | -1.280e+02 | 9.789e+12 |

Table 1: Summary statistics of the different datasets. "#examples" is the number of question/answer pairs in the dataset. avg-a, min-a and max-a are the mean, minimum and maximum of the ground truth answers in the datasets.

### 3.4 MOTIVATION

Our methodology centers on numerical reasoning tasks for three main reasons. First, this choice parallels the way overprecision is typically studied in cognitive science, where interval judgments on numerical questions provide a consistent and well-established measure of the bias (see Section 2.1). Second, numerical outputs offer a cleaner lens on model confidence than categorical or mixed formats, since they reduce the influence of linguistic biases such as positivity bias (Sumita et al., 2024). Third, unlike prior work that relied heavily on multiple-choice formats (Xiong et al.), we restrict our evaluation to direct question–answer pairs. This design avoids additional confounds introduced by MCQs—such as order effects and authoring artifacts (Sumita et al., 2024; Zheng et al.)—and allows us to isolate the model's behavior in producing interval estimates.

## 4 EXPERIMENTAL SETUP

**Datasets** We utilized the following datasets: FinQA (Chen et al., 2021), MedMCQA (Pal et al., 2022), MedQA (Jin et al., 2021), and MMLU (Hendrycks et al.). FinQA is designed for numerical reasoning over financial data. MedMCQA and MedQA are datasets consisting of medical multiple-choice questions (MCQs). MMLU is a versatile dataset that spans multiple domains, tasks, and topics. These datasets were selected to capture a range of numerical reasoning complexities. While MMLU focuses on general knowledge, FinQA and the medical datasets require more domain-specific expertise. FinQA, in particular, presents an additional level of difficulty as it involves reasoning directly from specialized financial reports of companies. We use numeric subsets of FinQA, MedQA/MedMCQA, and MMLU to span general knowledge and domain-specific reasoning; details of filtering to numeric single-answer form and scale statistics are provided in what follows.

**Data Processing** These datasets were filtered to extract questions with numerical answers that do not include units of measure, currency symbols, or any other strings conveying additional information about the number. In the case of FinQA, though it is designed for report-grounded reasoning, in this study, we evaluate models in a closed-book setting without attaching the source reports to stress tests of calibration under severe knowledge demands. Multiple-choice question (MCQ) data was converted to a direct answer format, ensuring that each question has a single answer without any options. Due to the limited number of numerical answers in the test splits of these datasets, we sampled questions from all splits during the process. Additionally, MedMCQA and MedQA were combined into a single dataset referred to as "Medical." Table 1 outlines the key characteristics of these datasets.

**Models** We focused on widely adopted black-box LLM models with established reliability, including GPT-3.5-turbo (Schulman et al., 2022) and GPT-4o-Mini (Achiam et al., 2023). The evaluation was deliberately restricted to highly-capable, instruction-adherent GPT-series models (GPT-3.5 and GPT-4) as a necessary methodological choice to establish a clean behavioral baseline for the overprecision phenomenon. This strategy serves a crucial robustness function by selecting architectures that are less prone to low-level numerical artifacts, such as "Benford's Curse" (Shao et al., 2025) or severe tokenization effects, prevalent in lower-capacity, open-source models (Singh & Strouse, 2024). By isolating the observed failure (insensitivity to confidence cues) as a higher-level cognitive or calibration deficit, we obtain a more robust signal.

**Study type and scope** Our goal is characterization, not competition. Using two widely adopted black-box models, we describe behavior under a fixed prompting/sampling protocol. We vary the self-refinement shot count and compare aggregation strategies purely to understand patterns; we do not select configurations to maximize any metric or to rank models. All figures (means ± s.d.) are descriptive.

## 4.1 PROTOCOL

**Phase 1 (Generation)**  Each question in the dataset is paired with a specific prompting strategy, sampling strategy, and confidence level ([60%, 70%, 80%, 90%, 95%]). These combinations are evaluated on an LLM over five trials to account for randomness. Each trial produces an interval with upper and lower bounds for the predicted answer.

**Phase 2 (Refinement)**  Answers generated in the first phase are refined using either aggregation or self-refinement strategies. For each question-answer pair, responses are sampled and processed through a refinement function to produce a new interval. To ensure cost efficiency, a single model is utilized throughout this phase. Two settings are considered: (1) Mixed confidence, where responses are sampled randomly across different confidence levels, and (2) Single confidence, where responses are sampled randomly within a specific confidence level.

For each combination, a single trial is randomly sampled, and evaluation metrics are computed over 10 runs. Both the mean and standard deviation are reported. Due to budget constraints, multiple prompts were not feasible for self-refinement; thus, a single trial per question-answer pair was used. This approach relies on prior experiments (i.e., the generation phase) to assume consistency in the results. The temperature for the models is set to 1 by default.

## 5 EVALUATION AND ANALYSIS

| dataset | model | P.S. | hit@95% mean | std | hit@90% mean | std | hit@80% mean | std | hit@70% mean | std | hit@60% mean | std | hit-avg mean | std | corr mean | std |
|---------|-------|------|------|-----|------|-----|------|-----|------|-----|------|-----|------|-----|------|-----|
| FinQA | gpt-3.5-turbo | vanilla | 6.16 | 0.24 | 5.50 | 0.23 | 6.47 | 0.30 | 6.79 | 0.20 | 7.42 | 0.28 | 6.47 | 0.09 | -0.0089 | 0.0070 |
| | | CoT | 7.04 | 0.25 | 7.16 | 0.36 | 7.33 | 0.49 | 7.35 | 0.34 | 7.55 | 0.21 | 7.29 | 0.17 | 0.0034 | 0.0143 |
| | gpt-4o-mini | vanilla | 21.14 | 0.35 | 18.95 | 0.41 | 18.25 | 0.36 | 16.05 | 0.43 | 17.04 | 0.45 | 18.29 | 0.20 | -0.0019 | 0.0038 |
| | | CoT | **21.54** | 0.41 | **20.29** | 0.51 | **20.32** | 0.43 | **19.05** | 0.41 | **19.75** | 0.46 | **20.19** | 0.12 | -0.0006 | 0.0089 |
| Medical | gpt-3.5-turbo | vanilla | 48.28 | 0.59 | 47.71 | 0.59 | 48.85 | 0.84 | 47.26 | 0.55 | 49.42 | 0.72 | 48.31 | 0.25 | -0.0051 | 0.0089 |
| | | CoT | 48.48 | 0.60 | 47.79 | 0.74 | 49.60 | 0.99 | 49.46 | 1.11 | 48.68 | 0.89 | 48.80 | 0.38 | -0.0004 | 0.0094 |
| | gpt-4o-mini | vanilla | 60.31 | 0.55 | 60.41 | 0.42 | 60.61 | 0.36 | 59.81 | 0.65 | 60.39 | 0.38 | 60.30 | 0.20 | 0.0097 | 0.0067 |
| | | CoT | **68.49** | 0.88 | **68.00** | 0.44 | **67.69** | 0.55 | **66.25** | 0.47 | **66.91** | 0.95 | **67.47** | 0.29 | 0.0119 | 0.0030 |
| MMLU | gpt-3.5-turbo | vanilla | 59.40 | 0.62 | 58.70 | 0.65 | 59.33 | 0.69 | 59.30 | 0.92 | 60.03 | 0.76 | 59.35 | 0.28 | 0.0030 | 0.0108 |
| | | CoT | 57.68 | 0.75 | 57.20 | 0.96 | 58.53 | 0.63 | 59.37 | 1.16 | 58.72 | 0.67 | 58.30 | 0.44 | -0.0068 | 0.0116 |
| | gpt-4o-mini | vanilla | 67.05 | 0.64 | 68.21 | 0.63 | 68.09 | 0.65 | 68.01 | 0.61 | 68.85 | 0.44 | 68.04 | 0.20 | -0.0052 | 0.0078 |
| | | CoT | **79.56** | 0.42 | **80.07** | 0.50 | **80.93** | 0.49 | **80.66** | 0.55 | **81.21** | 0.50 | **80.49** | 0.31 | 0.0019 | 0.0144 |

Table 2: Calibration evaluation in vanilla and CoT settings in phase 1 across two models and three datasets over 10 runs. We report the average and standard deviation across runs for each metric. The closer the hit rates are to their respective confidence level $c$, the better the calibration and vice versa. Additionally, a high correlation (corr) between confidence levels and predicted interval lengths reflects stronger self-confidence awareness in the LLM. P.S. refers to the prompting strategy. The results show a widespread miscalibration across datasets and models. CoT prompting has mixed effects (i.e. it didn't improve GPT-3.5-Turbo), which contradicts previous studies on overestimation (Xiong et al.).

## 5.1 LLMS ARE GENERALLY OVERPRECISE

Table 2 shows Phase 1 generation results. Across models and datasets, hit rates consistently fall far below target confidence levels, confirming systematic overprecision. At 95%, GPT-4o-mini covers only 21% on FinQA (–74 percentage points (pp)) and 60–68% on MedicalQA (–27 to –35 pp); even on MMLU, the strongest case, coverage is still 15–28 pp below nominal. These gaps demonstrate that models generate overly narrow intervals. CoT prompting improves GPT-4o-mini but has little effect on GPT-3.5-Turbo and slightly worsens its MMLU performance. This extends prior findings on LLM overestimation (Xiong et al.; Geng et al., 2024) to the domain of overprecision, though the lack of CoT benefits for GPT-3.5-Turbo contrasts with (Xiong et al.)'s results on categorical tasks. The counter-intuitive, mixed effects of CoT prompting suggest a fundamental disconnect between the generated reasoning trace and the model's internal uncertainty estimate. While CoT improves the quality of the point estimate, it fails to meaningfully engage the miscalibrated confidence mechanism, as the model may not genuinely understand the reasoning it produces (Wang et al., 2025). This issue is compounded by reasoning complexity: when the problem exceeds a critical compositional

| dataset | agg_strategy | hit-avg mean | std | hit@95% mean | std | hit@90% mean | std | hit@80% mean | std | hit@70% mean | std | hit@60% mean | std | corr mean | std |
|---|---|---|---|---|---|---|---|---|---|---|---|---|---|---|---|
| FinQA | LWA | 19.46 | 0.17 | 22.58 | 0.41 | 20.48 | 0.49 | 19.36 | 0.35 | 16.88 | 0.36 | 17.99 | 0.42 | -0.0013 | 0.0028 |
| | MIA | 18.74 | 0.14 | 21.84 | 0.34 | 19.44 | 0.26 | 18.87 | 0.36 | 16.49 | 0.32 | 17.09 | 0.39 | -0.0024 | 0.0022 |
| | Union | 33.88 | 0.16 | 35.87 | 0.48 | 34.54 | 0.38 | 34.44 | 0.28 | 31.89 | 0.33 | 32.64 | 0.32 | 0.0013 | 0.0021 |
| | iLWA | 17.01 | 0.19 | 19.29 | 0.28 | 17.71 | 0.52 | 17.05 | 0.38 | 15.17 | 0.44 | 15.83 | 0.36 | -0.0051 | 0.0018 |
| Medical | LWA | 56.03 | 0.18 | 55.88 | 0.57 | 55.48 | 0.47 | 56.05 | 0.59 | 55.99 | 0.42 | 56.76 | 0.54 | 0.0113 | 0.0036 |
| | MIA | 56.53 | 0.18 | 56.63 | 0.49 | 56.64 | 0.53 | 56.77 | 0.37 | 56.14 | 0.23 | 56.46 | 0.40 | 0.0133 | 0.0025 |
| | Union | 70.56 | 0.27 | 71.09 | 0.31 | 70.58 | 0.43 | 70.66 | 0.53 | 69.77 | 0.46 | 70.69 | 0.30 | 0.0129 | 0.0036 |
| | iLWA | 51.12 | 0.14 | 50.16 | 0.29 | 50.36 | 0.53 | 51.45 | 0.53 | 51.12 | 0.45 | 52.52 | 0.44 | 0.0127 | 0.0019 |
| MMLU | LWA | 58.39 | 0.13 | 56.17 | 0.56 | 55.59 | 0.58 | 58.31 | 0.58 | 59.87 | 0.51 | 62.00 | 0.34 | -0.0047 | 0.0083 |
| | MIA | 65.20 | 0.25 | 64.36 | 0.44 | 64.23 | 0.45 | 65.45 | 0.55 | 65.50 | 0.68 | 66.46 | 0.31 | -0.0032 | 0.0066 |
| | Union | 76.09 | 0.10 | 75.82 | 0.31 | 76.16 | 0.31 | 75.87 | 0.32 | 76.07 | 0.38 | 76.54 | 0.29 | -0.0019 | 0.0054 |
| | iLWA | 46.74 | 0.23 | 42.70 | 0.33 | 42.80 | 0.63 | 47.12 | 0.48 | 48.86 | 0.51 | 52.24 | 0.59 | 0.0007 | 0.0094 |

Table 3: Results of various aggregation-based refinement strategies on the GPT-4o-Mini model across different datasets in the single confidence setting, where sampling is performed separately for each confidence level. The results show that aggregation strategies generally don't improve overconfidence in LLMs in a single confidence setting except for the obvious Union strategy.

| dataset | CWA mean | std | LWA mean | std | MIA mean | std | Union mean | std | iLWA mean | std |
|---|---|---|---|---|---|---|---|---|---|---|
| FinQA | 19.58 | 0.37 | 21.47 | 0.34 | 19.23 | 0.28 | 55.04 | 0.38 | 16.02 | 0.27 |
| Medical | 52.84 | 0.48 | 54.90 | 0.43 | 53.18 | 0.42 | 81.78 | 0.18 | 44.88 | 0.39 |
| MMLU | 61.19 | 0.45 | 62.31 | 0.43 | 61.75 | 0.48 | 84.62 | 0.28 | 33.89 | 0.45 |

Table 4: Performance (average hit rate and its standard deviation) of various aggregation-based refinement strategies on the GPT-4o-Mini model across different datasets in the mixed confidence setting, with sampling conducted separately for each confidence level. The results show that aggregation strategies generally don't improve overconfidence in LLMs in a mixed confidence setting except for the obvious Union strategy.

threshold, the CoT trace and the model's accuracy can collapse (Shojaee et al., 2025), forcing the generation of an interval based on a flawed internal process.

### 5.2 LLMs' Confidence Does Not Correlate with Their Predictions

Table 2 shows that hit rates are flat across confidence levels and interval length barely correlates with imposed confidence, indicating that LLMs do not adjust intervals to explicit instructions. To probe further, we introduced Deviation Score (DS) and Interval Length Score (ILS). These reveal that interval size reflects task difficulty (FinQA > Medical > MMLU) rather than requested confidence. Appendix Figures 5 and 6 confirm dataset-level differences but, being averaged across confidence levels, do not establish stability across 60–95%. In short, LLMs widen intervals for harder domains but remain insensitive to confidence cues, unlike humans who at least attempt to widen intervals with higher confidence demands (Alpert & Raiffa, 1982; Soll & Klayman, 2004).

### 5.3 LLM Performance Is Affected by the Prompting Strategy, the Scale of the Answer, and the Task

Figure 2 demonstrates how the scale of ground truth answers influences LLM prediction accuracy. For example, in FinQA, predictions for answers near 0 tend to be more accurate, while accuracy declines for larger positive or negative values. Table 2 further emphasizes the impact of task type and prompting strategy on performance. Accuracy is significantly lower for specialized tasks such as FinQA and Medical, which require domain-specific knowledge, compared to general tasks like MMLU, which depend on broader knowledge without the need for specialized expertise.

### 5.4 Refinement affects precision
#### 5.4.1 Aggregation

To test robustness, we ran 10 simulations with random sampling (three trials per QA pair in the single-confidence setting; nine in the mixed-confidence setting) and report mean and standard deviation. Tables 3 and 4 present results for GPT-4o-Mini under vanilla prompting. While aggregation was

| dataset | kind | hit@95% | hit@90% | hit@80% | hit@70% | hit@60% | hit-avg | corr |
|---|---|---|---|---|---|---|---|---|
| FinQA | chosen | 20.56 | 18.42 | 17.73 | 16.33 | 17.36 | 18.08 | -0.0170 |
| | proposed | 16.91 | 15.75 | 15.00 | 13.26 | 13.46 | 14.88 | -0.0104 |
| Medical | chosen | 59.52 | 60.69 | 61.06 | 60.84 | 61.08 | 60.64 | 0.0191 |
| | proposed | 50.19 | 52.43 | 51.48 | 50.73 | 50.39 | 51.04 | 0.0062 |
| MMLU | chosen | 66.73 | 66.92 | 68.12 | 67.08 | 68.68 | 67.51 | 0.0021 |
| | proposed | 59.75 | 58.13 | 59.78 | 58.85 | 57.78 | 58.86 | 0.0030 |

Table 5: **Self-refinement in the single confidence setting**: Self-refinement of answers generated using vanilla prompts from the GPT-4o-Mini model across different datasets, utilizing the GPT-4o-Mini LLM. For each question-answer pair, three possible answers are sampled from each confidence level. "Chosen" refers to the answers selected by the LLM from the proposed options, while "Proposed" represents the new interval suggested by the LLM. Our self-refinement setup did not yield improvements, diverging from prior reports where self-probing improved calibration.

| dataset | kind | hit-avg |
|---|---|---|
| FinQA | chosen | 18.54 |
| | proposed | 15.56 |
| Medical | chosen | 60.59 |
| | proposed | 52.96 |
| MMLU | chosen | 65.61 |
| | proposed | 59.13 |

Table 6: **Self-refinement in the mixed confidence setting**: Using the GPT-4o-Mini model, self-refinement generates answers across datasets by sampling nine responses per question, regardless of confidence levels. "Chosen" refers to the LLM's selected answers, while "Proposed" represents the new intervals it suggests.

introduced as an exploratory baseline, the results confirm clear patterns: in the single-confidence case, LWA and MIA had mixed results, iLWA reduced performance, and union consistently delivered the largest coverage increases on MMLU and Medical by widening intervals—though this came with the expected cost to informativeness. Only the Medical dataset showed a slight increase in correlation between interval length and confidence. In the mixed-confidence case, union again produced the most reliable improvements, even exceeding its single-confidence performance, whereas other strategies fluctuated unpredictably across datasets.

### 5.4.2 SELF-REFINEMENT

Tables 5 and 6 show that self-refinement did not improve calibration over the vanilla setting, diverging from prior reports (Haran et al., 2010; Moore et al., 2015a; Xiong et al.). However, Appendix E shows that when models are given more candidate examples, their proposed intervals can modestly improve, though gains remain inconsistent and smaller than aggregation effects. To probe this discrepancy, we analyzed selection behavior during refinement (Appendix F) and found that models overwhelmingly favored the narrowest intervals—choosing the shortest option 40–60% of the time, far above chance—while rarely selecting the widest. This bias persisted even when narrow intervals excluded the true answer, with correct selections only modestly more balanced across intermediate widths. These results suggest that self-refinement operates as a narrowing heuristic favoring apparent precision, thereby reinforcing rather than mitigating overprecision.

## 6 DISCUSSION

This study offers key insights into LLM overprecision. Chain-of-thought (CoT) reasoning improved calibration for some models but not others, indicating model-dependent benefits tied to how well reasoning informs interval estimates. Calibration varied across tasks and answer scales, extending prior findings on categorical–numerical data (Xiong et al.). Refinement strategies had inconsistent effects: aggregation improved coverage mainly by widening intervals (union method), while others had limited or negative impact, differing from prior reports of consistent overestimation mitigation (Xiong et al.; Wen et al.). Self-refinement produced only marginal, non-generalizable gains (Appendix E), suggesting a narrowing bias that limits scalability. Overall, refinement success depended on dataset features, uncertainty type, and prompt design, highlighting the need for cross-domain evaluation. Miscalibration persisted via distinct mechanisms, confirming that overestimation and overprecision

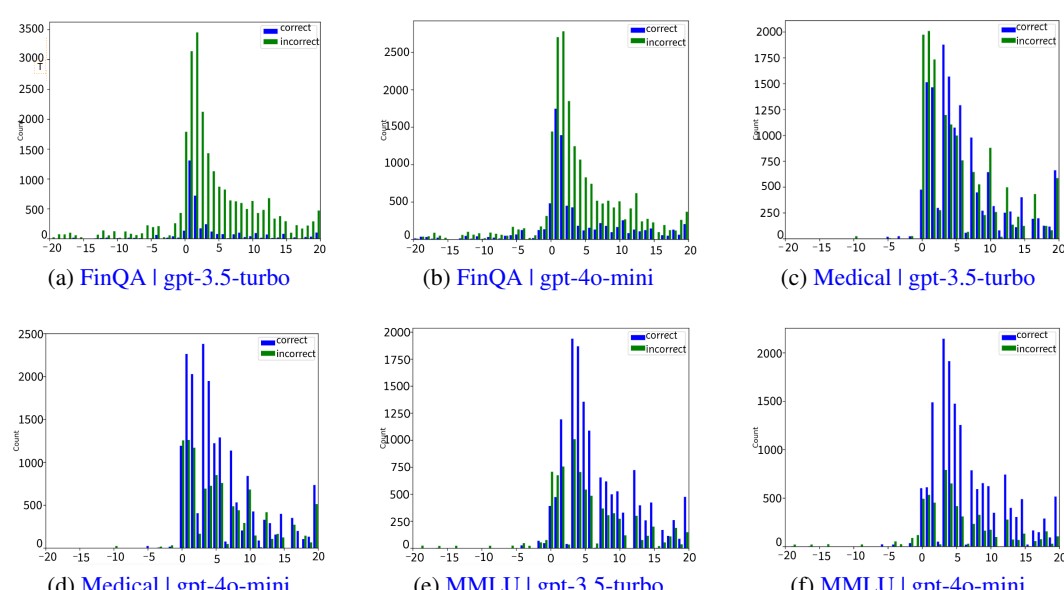

Figure 2: **Scale effect on calibration**: These figures show the distribution of the hit average for different answers in the vanilla prompt setting for different models on different datasets. The figures demonstrate that the performance is affected by the prompting strategy, the scale of the answer, and the task.

differ and require separate solutions. These results apply to the studied datasets and models; broader validation is needed to confirm generality.

## 7 CONCLUSION

This study explores the underexamined issue of overprecision in LLMs, revealing key behavioral limitations. LLMs show poor calibration in numerical tasks, with no clear link between interval length and confidence—indicating difficulty aligning interval size with explicit confidence cues. Calibration varies substantially by task, answer scale, and prompting technique. Refinement strategies prove largely ineffective, with self-refinement often reducing performance, contradicting prior results from cognitive science and general LLM research. Overall, within the tested domains and models, LLMs remain poorly calibrated for numerical reasoning and show limited sensitivity to confidence instructions. Overestimation and overprecision emerge as distinct phenomena, and broader cross-domain validation is required before generalizing these findings.

### LIMITATIONS

1) *Scope of Datasets*: This study primarily focused on two domains, finance and medicine, with some general knowledge tasks from MMLU. We believe this work can be further enhanced by extending experiments to other domains such as mathematics, law, biology, physics, and other fields involving numerical reasoning tasks.

2) *Scope of Models*: Due to budget constraints, we limited our experiments to two models. While these models exhibited varying behaviours, we aim to expand this study in the future by including a broader range of models to capture more diverse insights.

3) *Black-box setting*: The techniques proposed in this work are designed for black-box settings. However, we observed a lack of research on overprecision in white-box settings. Exploring this aspect could open new and interesting avenues for future research.

4) *Methodology*: Our prompting and refinement templates represent only one design that takes into consideration prior work from cognitive science and LLMs; alternative formulations may lead to different outcomes.

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

## A    CODE AND DATA

The code and dataset are available at `https://anonymous.4open.science/r/LLMOverprecision-2207`.

## B    GENERATION

**Prompting Strategy**    Let $\mathbf{p}_c$ represent a prompt parameterized by a confidence level $c$. This prompt includes a series of instructions that the LLM must follow to answer the question. These instructions can be divided into distinct sets. Formally, $\mathbf{p}_c$ can be expressed as:

$$\mathbf{p}_c(q_i) = [\text{GEN}, \text{CONF}_c, \text{CONFK}, \text{FORM}, \text{QUES}(q_i)] \tag{2}$$

where $[.]$ denotes text concatenation. Table 7 provides further details on the formulation and purpose of each instruction set. The initial prompt employs a vanilla prompting strategy. An alternative experimental variant utilizes the chain of thought (CoT) prompting strategy and is formulated as follows:

$$\mathbf{p}_c(q_i) = [\text{GEN}, \text{CONF}_c, \text{CONFK}, \text{FORM}, \text{CoT}, \text{QUES}(q_i)] \tag{3}$$

The formulation of CoT is in table 7.

**Sampling Strategy**    We employed a self-random sampling strategy that involves prompting the LLM multiple times to leverage the inherent randomness of the generation process. The prompts defined in Eqs. 2 and 3 are repeatedly fed to the LLM to obtain randomly sampled answers.

## C    REFINEMENT

We explore two complementary refinement strategies: (a) aggregation and (b) self-refinement. Aggregation combines multiple model-generated intervals into a single estimate. While aggregation is well established for categorical predictions, its application to interval outputs is less common. We therefore adapt several simple schemes (e.g., mean-, length-weighted, and union-based aggregation) to examine whether pooling interval estimates can improve coverage. Self-refinement, in contrast, asks the model to reconsider a set of its own intervals, select the most plausible one, and optionally propose a new interval. This design draws inspiration from cognitive science studies showing that access to peer judgments can reduce overprecision. In our setting, the model effectively plays the role of its own "peer reviewer," offering a first step toward testing whether reflective prompting can

| Instruction | Text | Objective |
|---|---|---|
| GEN | "please follow these instructions to ..." | General instructions that the LLM should follow |
| $CONF_c$ | "Please give us two numbers: a 'lower bound' and an 'upper bound'. The 'lower bound' is a number so low that there is only a $\frac{(100-c)}{2}\%$ probability that the right answer is less than that. Similarly, an 'upper bound' is a number so high that there is only a $\frac{(100-c)}{2}\%$ probability the right answer is more than that. In other words, you should be $c\%$ sure that the answer falls between the lower and upper bounds" | Instructing the LLM on the level of confidence that it should have in its answer. |
| CONFK | "The more unsure you are in your response the upper bound and the lower bound should be distant ..." | Giving the LLM general knowledge about confidence |
| FORM | "your answer should have the following format ..." | Formating instructions that facilitate the parsing of the LLM output |
| CoT | "give your step-by-step reasoning for why..." | Chain of Thought instructions that encourage the LLM to have better reasoning |
| $QUES(q_i)$ | "Question: $[q_i]$" | The question that the LLM should answer. |

Table 7: Sets of instructions that are used in the prompts. 'instruction' represents the abbreviation used in the paper for a particular set of instructions. 'Text' is the instruction text. 'objective' is the purpose of having that set of instructions.

mitigate interval miscalibration. We note that these strategies represent only one set of possible designs. Alternative aggregation rules or self-refinement prompts may yield different outcomes, and we view our implementations as exploratory baselines rather than definitive solutions.

## C.1 AGGREGATION STRATEGIES

Let $[x_i, y_i]_{i=1}^N$ represent a set of $N$ intervals obtained by prompting the LLM $N$ times using variants of the previously discussed prompts. Let $c_i$ denote the confidence level imposed on the LLM in the prompt to generate the $i$th answer. Interval aggregation combines the upper and lower bounds of these output intervals to produce an aggregated interval. Formally, this strategy can be defined as follows:

$$X = \frac{\sum_{i=1}^N w_i x_i}{\sum_{j=1}^N w_i}, \qquad Y = \frac{\sum_{i=1}^N w_i y_i}{\sum_{j=1}^N w_i} \tag{4}$$

where $X$ and $Y$ are the lower and upper bounds of the aggregated interval, respectively, and $w_i$ is a weight that determines the contribution of the $i$th interval to the overall aggregation. The values of the $w_i$'s are determined based on various weighting schemes. In this study, we utilized the following:

- Mean interval aggregation (MIA): This strategy gives each interval equal weighting as follows: $w_i = 1, \forall i$.

- Length weighted aggregation (LWA): This strategy weighs longer intervals more than smaller intervals as follows: $w_i = d_i, \forall i$, where $d_i = y_i - x_i, \forall i$.

- Inverse length weighted aggregation (iLWA): This strategy weighs shorter intervals more than longer intervals as follows: $w_i = \bar{d}_i, \forall$, where $\bar{d}_i = \frac{1}{y_i - x_i}, \forall i$.

- Confidence weighted aggregation (CWA): in cases where the same query is prompted at different confidence levels, confidence intervals can be used to weigh the intervals as follows: $w_i = c_i, \forall i$.

| prompt $\mathbf{p}^{\text{refine}}([x_i, y_i, c_i]_{i=1}^N, q, e))$ |
|---|
| - Context: A group of people were given a question |
| ... |
| - Instructions: |
| - Analyse the question, the answers to the question and their corresponding confidence level. |
| - Determine the most likely ... |
| - give your reasoning ... |
| - Your output should have the following format ...: |
| {     "chosen_answer":[lower_bound,    upper_bound],    "chosen_reason":,    "proposed_answer":[lower_bound,   upper_bound], "proposed_reason": } |
| - Question: $q$ |
| - Possible Answers: |
| $e$ examples $= \begin{cases} x_i \lvert y_i \rvert c_i \\ \cdots \\ x_j \lvert y_j \rvert c_j \end{cases}$ |

Table 8: Self-refinement prompt. The prompt takes as inputs a question $q$ and a set of $e$ potential answers from the generation phase.

In addition to the previous schemes, we also experiment with the union of intervals (Union), which can be presented formally as follows:

$$X = \min(\{x_i\}_i), \qquad Y = \max(\{y_i\}_i) \tag{5}$$

## C.2 SELF-REFINEMENT

For a set of $N$ responses and their corresponding confidence levels, $A = [x_i, y_i, c_i]_{i=1}^N$, obtained during the generation step for a question $q$, self-refinement involves improving the LLM's responses by prompting it to evaluate the initial answers, select the most probable one, and propose an enhanced response. This process takes into account the confidence levels associated with each answer generated in the initial step. Formally, this process can be expressed as follows:

$$\text{LLM}(\mathbf{p}^{\text{refine}}([x_i, y_i, c_i]_{i=1}^N, q, e)) = \begin{cases} (X^{\text{old}}, Y^{\text{old}}) \\ (X^{\text{new}}, Y^{\text{new}}) \end{cases} \tag{6}$$

where $X^{\text{old}} \in \{x_i\}_i$ and $Y^{\text{old}} \in \{y_i\}_i$ are bounds from the existing list of proposed bounds within which the potential answer may lie; $X^{\text{new}}$ and $Y^{\text{new}}$ represent the new lower and upper bounds, respectively, generated by the LLM based on the potential answers and their associated confidence levels; and $e$ denotes the number of elements sampled from $A$. Table 8 provides a summary of the formulation of the self-refine prompt.

## D EVALUATION METRICS

We evaluate the LLM on two primary tasks: (a) precision calibration and (b) confidence understanding. Let $\hat{A}^c = (q_i, a_i, [x_i^c, y_i^c])_i$ represent a set of questions $q_i$ with their corresponding ground truth answers $a_i$ and the LLM-generated intervals $[xi^c, y_i^c]$ at a confidence level $c$, obtained using a variation of the previously discussed prompting techniques. In line with existing literature on overprecision in cognitive science (Soll & Klayman, 2004; Moore et al., 2015a), we use the hit metric, which calculates the percentage of instances where the ground truth answers fall within the generated intervals. Formally, this can be expressed as follows:

$$\text{hit@}c\% = \frac{1}{|\hat{A}^c|} \sum_{i=1}^{|\hat{A}^c|} I(a_i \in [x_i^c, y_i^c]) \tag{7}$$

where $I$ is the indicator function, defined as $I(\text{cond}) = 1$ if the condition cond is satisfied, and $I(\text{cond}) = 0$ otherwise. Additionally, we compute Pearson's correlation coefficient (Sedgwick, 2012) between the confidence levels and the lengths of the intervals to assess the LLM's awareness of its own self-confidence (Moore & Healy, 2008).

To study trends and variations in interval size and the deviation from the interval, we introduce two metrics: a) deviation score (DS) and b) interval length score (ILS). The DS measures the amount that the interval deviates from the expected answer, and the ILS measures how large the predicted interval is. DS can be expressed as follows:

$$\text{DS}^c = \frac{1}{|\hat{A}^c|} \sum_{i=1}^{|\hat{A}^c|} \left( \frac{\max(m_i^c, 0)}{|m_i^c| + 1} \right)^2 \tag{8}$$

with $m_i^c = \max(x_i^c - a_i, a_i - y_i^c)$. This metric equals 0 if the expected answer is in the predicted interval otherwise, the further the answer is from the interval, the higher the score. The ILS metric can be expressed as follows:

$$\text{ILS}^c = \frac{1}{|\hat{A}^c|} \sum_{i=1}^{|\hat{A}^c|} \frac{y_i^c - x_i^c}{\max(|y_i^c|, |x_i^c|)} \tag{9}$$

This metric considers the interval's length and the values' scale to penalize larger intervals with lower scales more than smaller intervals with larger scales.

## E    EFFECTS OF NUMBER OF POSSIBLE ANSWERS ON SELF-REFINEMENT

Figure 4 shows how the performance of GPT-4o-mini in the self-refinement process as a function of the number of provided examples for different datasets in different settings. The "chosen" answers performance is not consistent across datasets and settings. However, the accuracy of the proposed responses generally increases with the number of examples in most settings and datasets (except MMLU in the "single" setting). The general trend of improved accuracy with an increasing number of examples suggests that the model benefits from seeing more context or task-specific information during the self-refinement process. This aligns with the principle that additional examples provide more opportunities for the model to learn patterns or clarify ambiguities, especially in few-shot learning setups. While proposed intervals improve slightly as the number of examples increases, the effect remains limited and does not resolve the broader pattern of miscalibration observed in the main results.

## F    CHARACTERISTICS OF INTERVALS SELECTED DURING SELF-REFINEMENT

To analyze why self-refinement fails to improve calibration, we examined which interval lengths models select when given four candidate intervals ranked from shortest (1) to longest (4). Figure 3 shows distributions across tasks, aggregated over all cases (left), stratified by whether the chosen interval contained the true answer (middle), or missed it (right).

Across all datasets, models exhibit a strong bias toward the shortest intervals: rank 1 was chosen in more than 40–60% of cases, far above the 25% expected under uniform choice. By contrast, the widest interval (rank 4) was almost never selected. Models exhibit a persistent bias toward the narrowest intervals. In the incorrect stratum, the shortest option (rank 1) remains the most frequently selected across all datasets, whereas the widest (rank 4) is rarely chosen. In the correct stratum, selections shift slightly toward longer intervals (ranks 2–3), but the shortest interval still accounts for the largest share. Thus, self-refinement favors apparent precision over coverage, and any accuracy gains largely arise when a narrow interval happens to include the truth, not because the model strategically widens its choice.

This pattern indicates that self-refinement operates more as a narrowing heuristic—favoring seemingly precise intervals—rather than as a calibration mechanism. As a result, refinement systematically reinforces overprecision instead of mitigating it.

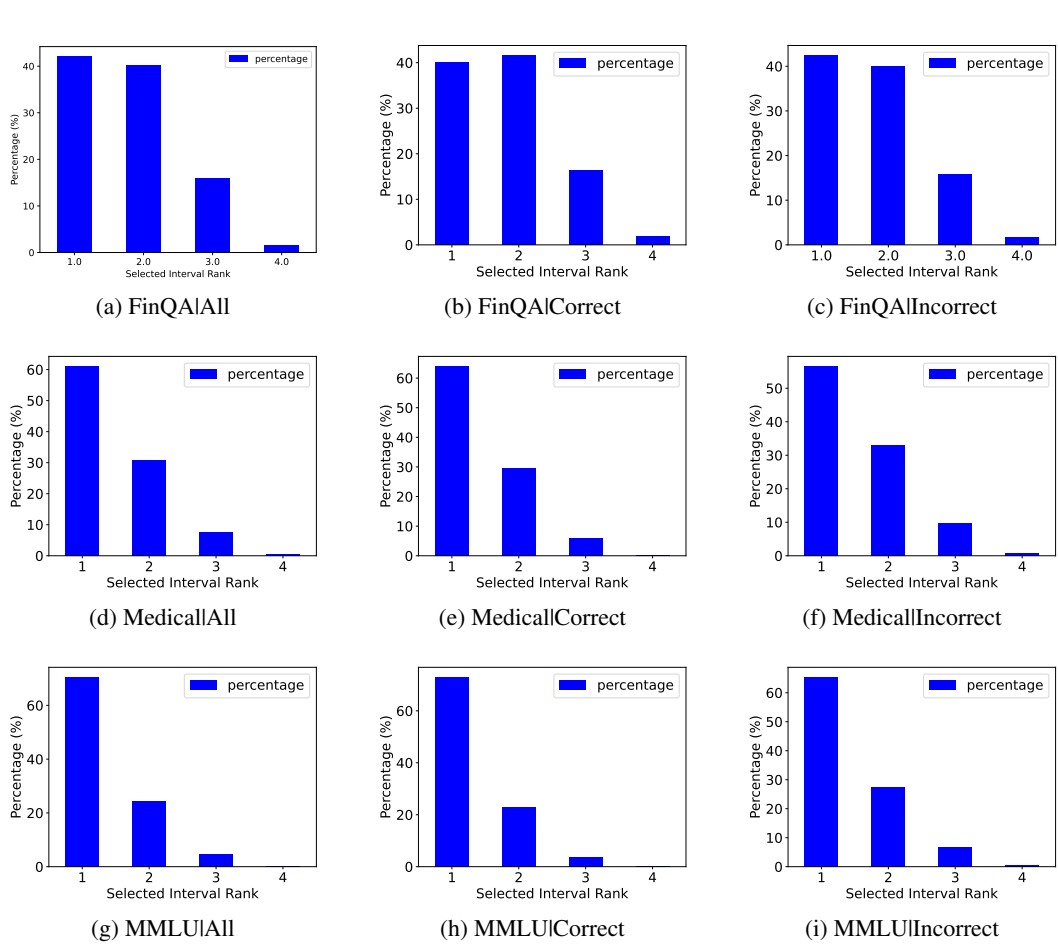

Figure 3: Distribution of selected intervals ranked by width (1=shortest, 4=longest). Left: all chosen answers. Middle: cases where the chosen interval contained the correct answer. Right: cases where it did not. Models overwhelmingly favor the narrowest intervals, even when these miss the ground truth, highlighting a bias toward overprecision.

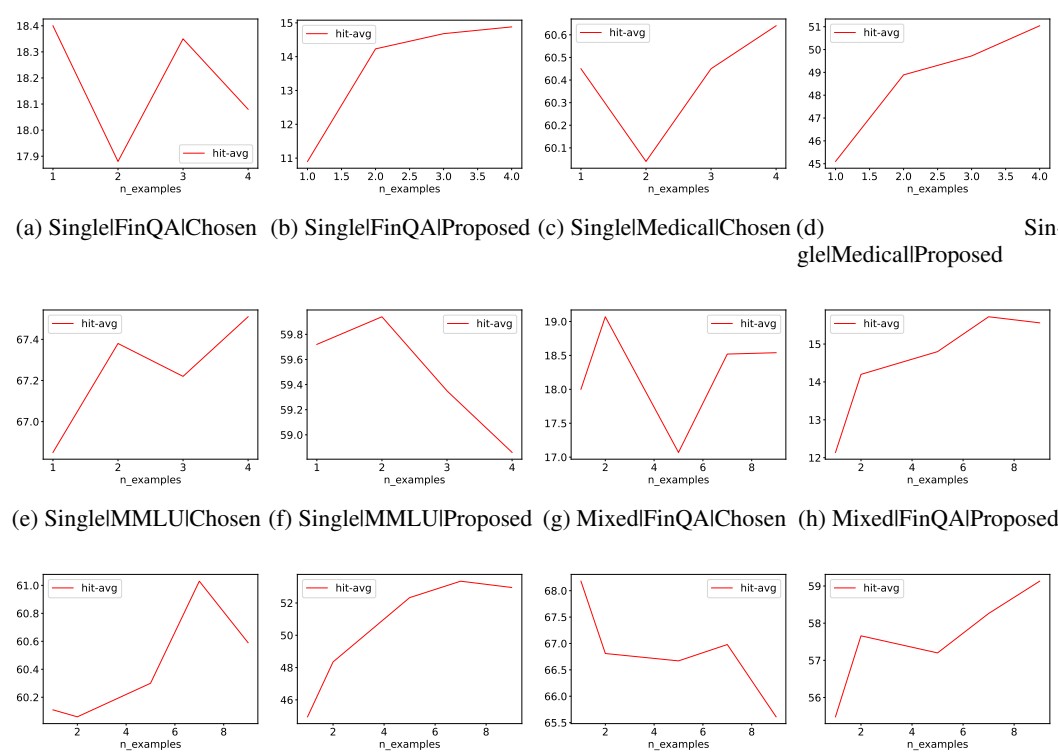

(a) Single|FinQA|Chosen  (b) Single|FinQA|Proposed  (c) Single|Medical|Chosen  (d) Single|Medical|Proposed

(e) Single|MMLU|Chosen  (f) Single|MMLU|Proposed  (g) Mixed|FinQA|Chosen  (h) Mixed|FinQA|Proposed

(i) Mixed|Medical|Chosen  (j) Mixed|Medical|Proposed  (k) Mixed|MMLU|Chosen  (l) Mixed|MMLU|Proposed

Figure 4: The hit average metric as a function of the number of examples provided in the self-refinement prompt. The titles of the subfigures are organized as follows: [setting][dataset][kind]. The setting can either be Single or mixed (refer to the experimental protocol for more detail). The kind can either be "chosen" for answers that were selected by the LLM to be the most correct. The kind can also be "proposed" for the answers that were proposed by the LLM but didn't exist in the provided examples.

# G  THE EFFECTS OF DIFFERENT EXPERIMENTAL SETTINGS ON THE LENGTH AND DEVIATION OF THE INTERVALS

In this section, we study the effects of different datasets, models and prompting techniques on the length and deviation of the intervals. Figures 5 and 6 show the distributions of the average DS and ILS metrics for all confidence levels, respectively, in various experimental settings.

Figure 5 shows that the deviation scores are lower in MMLU relative to Medical dataset, which in turn has lower scores than those of the FinQA dataset. This reinforces the results shown in tables 2 and 3 and the findings in section 5, and demonstrates that those trends are not produced by outliers, but are consistent across each dataset.

Figure 6 shows that the average lengths of the intervals in FinQA dataset are larger than those of Medical dataset, which also has intervals larger than the MMLU dataset. This demonstrates that an LLM varies its interval size depending on how certain it is of the answer, which in addition to the previous findings about the lack of correlation between the confidence level and interval size, shows that LLMs can't adjust their confidence following instructions but they are still aware at a certain level of the task hardness and their lack of knowledge.

The effects of the different choices of prompting techniques and LLMs on the ILS and DS are mixed. In some cases, GPT-4o-mini significantly improved the ILS and DS over GPT-3.5-Turbo, and in some cases, the effect of model change is negligible or reversed. The same can be said for prompting techniques.

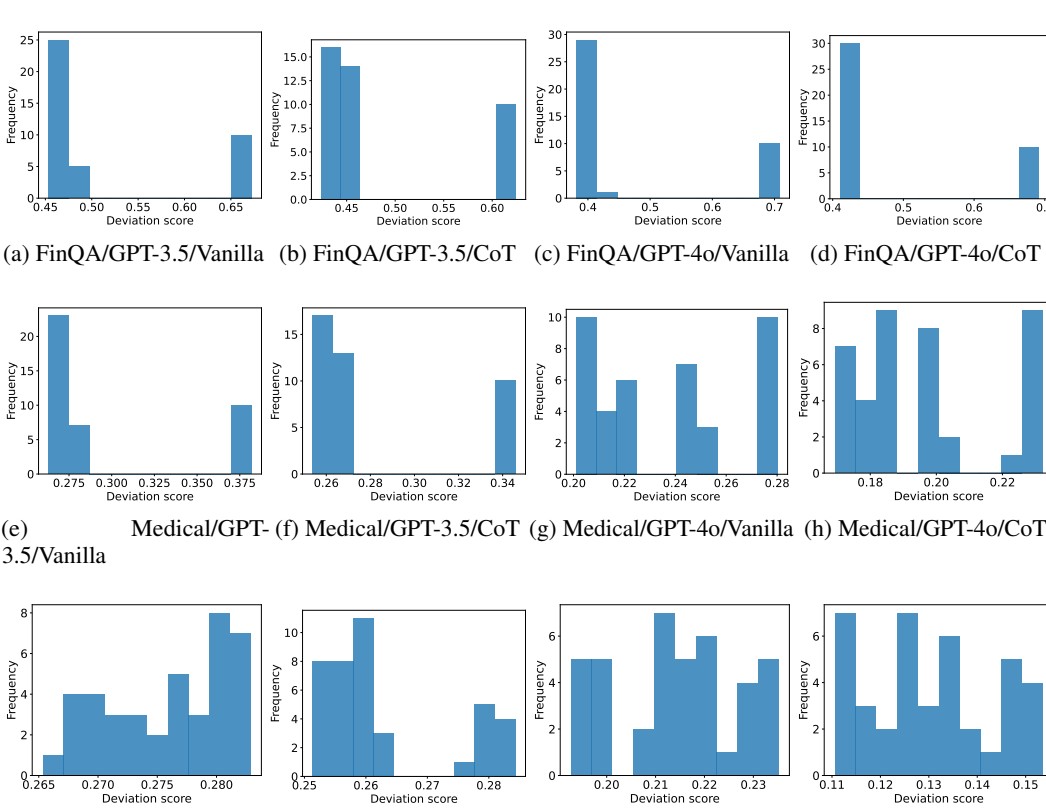

Figure 5: The figures show the distribution of the average DS metric across confidence levels for different datasets, in different models for vanilla and CoT prompts. GPT-3.5 is short for GPT-3.5-turbo, and GPT-4o is short for GPT-4o-mini. The DS values are lowest for MMLU, higher for the Medical dataset, and highest for FinQA. This supports earlier results in Tables 2 and 3 and Section 5, confirming that the observed trends are consistent across datasets and not driven by outliers.

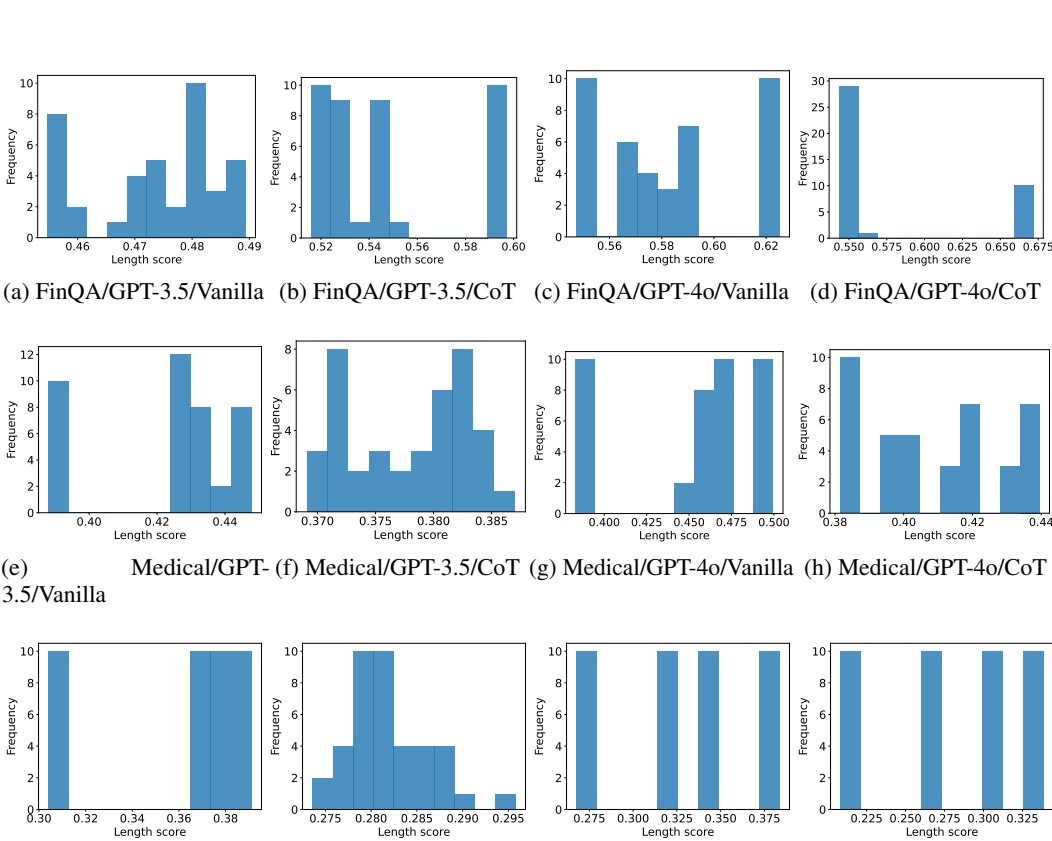

(a) FinQA/GPT-3.5/Vanilla  (b) FinQA/GPT-3.5/CoT  (c) FinQA/GPT-4o/Vanilla  (d) FinQA/GPT-4o/CoT

(e) Medical/GPT-3.5/Vanilla  (f) Medical/GPT-3.5/CoT  (g) Medical/GPT-4o/Vanilla  (h) Medical/GPT-4o/CoT

(i) MMLU/GPT-3.5/Vanilla  (j) MMLU/GPT-3.5/CoT  (k) MMLU/GPT-4o/Vanilla  (l) MMLU/GPT-4o/CoT

Figure 6: The figures show the distribution of the average ILS metric across confidence levels for different datasets, in different models for vanilla and CoT prompts. GPT-3.5 is short for GPT-3.5-turbo, and GPT-4o is short for GPT-4o-mini. The figures show that interval lengths are largest in FinQA, followed by Medical, and smallest in MMLU. This suggests that LLMs adjust interval size based on task difficulty, reflecting an awareness of uncertainty. However, combined with earlier findings on the lack of correlation between confidence and interval size, it indicates that while LLMs sense task hardness, they struggle to align their confidence with explicit instructions.

