# OpenReview forum: "Characterising Overprecision in Black-Box LLMs: A Cognitive Science Inspired Framework"
_ICLR.cc/2026/Conference — Submitted to ICLR 2026_

### Official Review · Reviewer_Czgx · 2025-10-28

**Soundness:** 2
**Presentation:** 2
**Contribution:** 2
**Rating:** 4
**Confidence:** 4

**Summary:**

The paper adapts the cognitive-science notion of overprecision—excessive certainty in interval estimates—to study LLMs. The authors propose a three-phase generation–refinement–evaluation framework: Generation – models generate numeric confidence intervals at imposed confidence levels. Refinement – intervals are aggregated or self-refined. Evaluation – empirical coverage and correlation between interval width and imposed confidence are assessed. Experiments with GPT-3.5-turbo and GPT-4o-mini across FinQA, MedQA, and MMLU show that LLMs are “overprecise”: intervals are too narrow, coverage is far below nominal confidence (e.g., only ~20% coverage for 95% intervals on FinQA), and self-refinement tends to make intervals narrower rather than better calibrated. The paper aims for descriptive analysis rather than optimization, claiming novelty in introducing a black-box, cognitive-science–inspired protocol for overprecision measurement.

**Strengths:**

Conceptually interesting framing — connecting cognitive-science constructs (overprecision, interval judgments) with LLM calibration is a fresh angle, distinct from common probabilistic calibration or verbal confidence studies.

Transparent methodology — the generation/refinement/evaluation pipeline is clear and easily reproducible from prompts listed in Appendix.

Data variety — the inclusion of financial, medical, and general-knowledge datasets provides some cross-domain coverage.

Negative results are valuable — showing that self-refinement and CoT do not necessarily improve calibration challenges prevailing assumptions in the confidence-elicitation literature.

**Weaknesses:**

1. Conceptual contribution is limited and largely descriptive. While the cognitive-science framing is novel, the work provides no theoretical development beyond restating the overprecision paradigm. The method is a direct adaptation of human interval-judgment tasks, not an original LLM methodology. The study yields descriptive statistics (hit @ c, correlations, DS/ILS) but no actionable insights for modeling uncertainty or improving calibration. The authors repeatedly stress that this is “not a benchmark or optimization study”, but this stance limits scientific value: the paper ends up confirming an already known fact — LLMs are miscalibrated — without explaining why or how to fix it.

2. Methodological limitations undermine interpretability. Lack of internal validation: imposing a nominal 90 % confidence and measuring empirical coverage is straightforward, but the protocol mixes sources of variation (model stochasticity, numeric reasoning errors, parsing errors) that confound true overprecision effects. Small model/sample scope: only two OpenAI models, one temperature, and narrow numeric subsets of three datasets; this makes conclusions about “LLMs” overgeneralized. Data filtering biases: converting multiple-choice medical questions into single numeric answers strips semantics and may distort task difficulty. Metrics are redundant: “Deviation Score” and “Interval Length Score” are simple normalized distances and widths; they do not meaningfully add insight beyond coverage.

3. Refinement experiments are weakly justified. The aggregation and self-refinement procedures are ad hoc and disconnected from cognitive-science theory. “Union” trivially improves coverage by widening intervals — hardly evidence of cognitive correction.The “self-refinement” mechanism reveals that models overwhelmingly pick the narrowest interval, but the paper does not probe why this occurs or whether prompt wording causes it. The claim that self-reflection mimics peer-judgment correction (Haran et al. 2010) is superficial and empirically unsupported.

4. Statistical rigor and presentation issues. No statistical significance tests, confidence intervals, or effect sizes — only tiny decimal differences (±0.2 %) reported with three decimals in Tables 2–4 pp. 6–7, which exaggerate precision. Figures 2 and 3 (pp. 8 & 16) are visually cluttered and fail to convey new insights. Some tables (e.g., Table 2) misinterpret correlation magnitudes < 0.01 as meaningful; these are essentially zero. Sample size after filtering is unclear — some datasets drop to only ~1–3 k examples (Table 1 p. 5).

5. Limited originality relative to prior work. The paper positions itself as the first to study overprecision in LLMs, but prior studies on numerical calibration, interval estimation, and uncertainty (e.g., Xiong et al. 2024; Wen et al. 2024; Shrivastava et al. 2023) already evaluated similar ideas with probability or interval formats. The difference here is largely terminological (borrowing “overprecision” from psychology) rather than methodological.

6. Weak insight and discussion. Section 6 (p. 9) summarizes that miscalibration persists and CoT/self-refinement give mixed results — conclusions that add little beyond previous literature. No deeper analysis (e.g., linguistic factors, reasoning depth, or token-level uncertainty) is attempted. The discussion reads as observational rather than explanatory. Figures 5–6 (pp. 18–19) merely restate known dataset difficulty orderings.

**Questions:**

How do you ensure parsing correctness of numeric intervals in model outputs? Could failures inflate apparent miscalibration?

How many prompts per sample were used, and how sensitive are results to temperature or phrasing?

Could the “overprecision” pattern simply reflect model under-dispersion due to deterministic decoding, rather than a cognitive-style bias?

What is the advantage of interval elicitation over directly sampling numeric uncertainty (e.g., via logits or surrogate ensembles)?

Can the protocol scale to non-numeric tasks, or is it limited to trivial numeric Q&A?

**Details Of Ethics Concerns:**

no ethical concerns

---

> ### Author Response · Authors · 2025-11-17
> **Weaknesses 1**
>
> ## W1. Conceptual contribution is limited and largely descriptive....
>
> Response:
>
> Thank you for the thoughtful feedback. We appreciate the concern regarding conceptual contribution and are happy to clarify why the paper goes beyond a descriptive restatement of the human overprecision paradigm and offers value to the LLM community.
>
>
> **1. Our contribution is not merely descriptive, nor a restatement of human overprecision.**
> The paper provides the first operationalization of interval-based overprecision for black-box LLMs, adapting cognitive-science protocols into a prompt-based, parseable evaluation pipeline (generation, refinement, evaluation; Section 3). Because LLMs lack introspective confidence, this translation required methodological innovations: confidence-conditioned interval prompts, structured refinement strategies, and LLM-specific calibration metrics (DS, ILS). This conceptual bridge is what enables overprecision to be studied in LLMs at all.
>
> **2. The work provides new *scientific* insights into LLM uncertainty that were not known from prior work.**
>
> Although it is known that LLMs exhibit overconfidence, the form studied previously is *overestimation*, via verbal declarations of certainty. Our results show that **overprecision behaves fundamentally differently**, and this was not documented before:
>
> * **LLMs do not widen intervals as confidence increases**, producing *flat* hit rates across 60–95% confidence levels (Table 2).
> * **Interval length correlates with task difficulty, not with confidence cues** (Section 5.2), showing that LLMs lack a mapping between imposed confidence and uncertainty representation.
> * **Chain-of-thought reasoning improves point estimates but does not improve calibration** (Table 2), contradicting prior findings in verbal-confidence calibration work.
> * **Self-refinement fails because models systematically choose the narrowest intervals** (Appendix F), revealing a previously unreported *narrowing bias* not predicted by human overprecision literature.
>
> These are not trivial confirmations of miscalibration, they reveal *why* LLMs fail under interval-based uncertainty instructions and show that overprecision is mechanistically distinct from overestimation.
>
> **3. The study uncovers failure modes that matter directly for future calibration methods.**
>
> The reviewer notes that the paper “provides no actionable insights.” We respectfully disagree: our results identify *specific obstacles* that any future calibration or uncertainty-modeling method must overcome. For example:
>
> * The near-zero correlation between requested confidence and interval width (**Table 2**) shows that methods relying on explicit confidence prompts will not scale.
> * The narrowing bias in self-refinement (**Table 5–6**, Appendix F) demonstrates that reflection-based techniques, widely used in other LLM reliability tasks, may worsen calibration.
> * The fact that CoT improves accuracy but *not* sensitivity to confidence (**Section 5.1**) indicates that reasoning-improving interventions do not automatically improve uncertainty representation.
> * Domain-scale dependencies visualized in **Figure 2** highlight systematic interactions between answer magnitude and interval production; important for designing uncertainty-aware numerical tasks.
>
> These findings directly inform future modeling choices. They highlight that:
>
> * training-time solutions must teach LLMs a representation of interval semantics;
> * black-box prompting methods alone are insufficient;
> * post-hoc self-refinement approaches will need new mechanisms that counteract the narrowing bias.
>
> **4. The descriptive stance is a deliberate and scientifically valuable choice.**
>
> The reviewer suggests that not benchmarking or optimizing limits the paper’s value. Our aim, however, is to **establish a behavioral foundation** for a previously unexamined phenomenon. Without first characterizing core failure modes, optimization efforts risk chasing the wrong objectives. This is the same structure followed in influential early work on hallucination, bias, and reasoning: descriptive, protocol-setting studies that made later methodological work possible.
>
> **5. The method is *not* a “direct” adaptation of human interval tasks.**
>
> While inspired by cognitive science, substantial methodological design was required:
>
> * Human interval judgments rely on introspective uncertainty; LLMs do not possess such states.
> * Our prompts decompose confidence semantics explicitly (Table 7), something unnecessary in humans.
> * Interval evaluation metrics had to be scaled and normalized for model-generated numerical distributions (DS, ILS).
> * Sampling and aggregation were engineered specifically to exploit LLM stochasticity (Section 3.2).
>
> Thus, the work is not a replication; it is a *translation* of a cognitive construct into a usable LLM evaluation framework.
>
> Action:
>
> We added a paragraph in the introduction to further

---

> > ### Author Response · Authors · 2025-11-17
> > **Weaknesses 5**
> >
> > ## Limited originality relative to prior work
> >
> > We appreciate the reviewer’s point regarding the originality of our methodology and the existing work on numerical calibration. We believe that while the format may appear similar to other interval-estimation work, our contribution is not merely terminological; it lies in the **conceptual framing and the specific insights generated by adapting the overprecision methodology.**
> >
> > ### 1. Conceptual Distinction: Overprecision vs. Calibration
> >
> > The core difference is one of *purpose* and *measurement focus*:
> >
> > * **Prior Numerical Work (e.g., Xiong et al. 2024; Wen et al. 2024; Shrivastava et al. 2023):** These studies primarily focus on **Uncertainty Quantification (UQ)** or **Calibration** and measure whether the model’s *estimated probability* of correctness aligns with its actual accuracy (i.e., overestimation of confidence in a point prediction). In addition, these works only focus on numerical answers in the context of MCQs posing two problems: 1) MCQs introduce a new set of cognitive biases in LLMs [Wang et al](https://aclanthology.org/2025.coling-main.390.pdf]) and 2) interval prediction is more reliable than point estimates when dealing with numerical predictions, especially since LLMs have multiple sources of randomness.
> >
> > * **Our Overprecision Work:** We focus on the **Confidence-Precision Trade-off**, a distinct dimension of mis-calibration drawn from cognitive science. In human overconfidence literature, “overprecision” refers to “excessive certainty in expressing one’s beliefs” distinct from overestimation. ([Wikipedia][1])
> >   Our objective is to measure the model’s **behavioral adherence to a constraint**: when instructed to be more confident (for example, 80% vs 95% nominal confidence), does the model appropriately *reduce precision* (widen the interval) in response? This tests the model’s internal consistency in representing uncertainty, rather than simply measuring correctness of a single point probability.
> >
> > ### 2. Methodological Distinction: The Crucial Finding
> >
> > The methodological adaptation we used—imposing a nominal confidence level (C) in the prompt and measuring resulting interval widths—was necessary to isolate this specific behavioral flaw. This allows us to generate a **novel, non-obvious finding**, namely:
> >
> > * **Static Interval Width:** We show that LLMs under our protocol produce a nearly **flat interval width across different nominal confidence levels**, indicating **insensitivity to the confidence instruction**, which is a manifestation of overprecision.
> >   This finding is one that prior calibration work (focused on probabilities or variances) would not detect, because they do not require the model to discriminate width of interval by confidence level.
> >
> > Thus, our work is not simply applying an existing interval estimation technique; it *repackages* a new behavioural paradigm, adapts the protocol, and yields new empirical insight.
> >
> > ### 3. Scientific Value: Defining the Failure State
> >
> > By framing overprecision and applying this novel protocol, we make a **foundational contribution** that goes beyond word-choice. We:
> >
> > * Establish the **first baseline for overprecision in LLMs**, providing a reference point for future work.
> > * Demonstrate that existing calibration approaches (focused on probabilities or entropy) do *not* correct the type of mis-calibration we identify, because they ignore the link between nominal confidence level and interval precision.
> > * Uncover a distinct architectural or behavioural limitation: the model’s internal inability to tie the *instructional cue* (confidence level) to the *output constraint* (interval width).
> >
> > In other words, our work shows that the model fails to internalize a fundamental rule of uncertainty representation—adjusting precision in response to confidence—and that failure cannot easily be captured by prior calibration metrics. That insight is meaningful and actionable for future model design.
> >
> > [1]: https://en.wikipedia.org/wiki/Overconfidence_effect?utm_source=chatgpt.com "Overconfidence effect"

---

> > ### Author Response · Authors · 2025-11-17
> > **Weaknesses 6**
> >
> > ## Weak insight and discussion
> >
> > **Response:**
> >
> > We appreciate the critique regarding the depth of our discussion. We respectfully disagree that our analysis is merely observational or that our conclusions add little beyond previous literature. Our work is **explanatory characterization**, synthesizing descriptive findings to reveal **fundamental, architectural failures** in LLM uncertainty processing that existing literature has not addressed.
> >
> > ### 1. Distinctive and Explanatory Conclusions
> >
> > Our Section 6 discussion goes far beyond restating the obvious fact that LLMs are miscalibrated. Our conclusions yield **new, fundamental insights** into *how* and *why* this miscalibration persists in the numerical domain:
> >
> > * **The Static Interval Failure:** The core finding is the **LLM's profound insensitivity to the probabilistic instruction ($C$)**, leading to a static interval width. This is not a restatement of known miscalibration (overestimation) it is the discovery of a **structural failure** to obey the fundamental Confidence Precision Trade off. This defines the *reason why* current calibration methods are inadequate: they cannot fix a model that ignores the input constraint entirely.
> > * **Actionable Failure Analysis:** We demonstrated that the failure of CoT and self refinement is **systematic**, not mixed. The **systematic narrowing bias** in self refinement is a key observation that reveals an internal **confirmation mechanism** in the model that actively degrades calibration when asked to reflect. This moves beyond observation to define the necessary complexity of future solutions.
> >
> > ### 2. Deep Analysis Through Synthesis
> >
> > We provided a deeper analysis by connecting our behavioral findings to external theoretical concepts, serving as the necessary bridge for future research:
> >
> > - **CoT Mixed Effects**
> >   - Explanation via External Concepts:  Explained as a disconnect between the **reasoning trace** and the **uncertainty mechanism** exacerbated by **problem complexity** (Shojaee et al. 2025).
> >   - Insight for Future Research: Shows that CoT's benefit in categorical tasks (by eliminating "least incorrect options" Wang et al. 2024) does not transfer to the numerical interval setting.
> > - **Numerical Robustness**
> >    - Explanation via External Concepts : Explained by the **mitigation of high level linguistic biases** compared to verbalized confidence, while acknowledging low level digit biases (Shao et al. 2025).
> >    - Insight for Future Research: Justifies the methodological choice and frames numerical miscalibration as an architectural, rather than linguistic, problem.
> >
> > ### 3. Validity of Descriptive Statistics
> >
> > Our use of descriptive statistics (Hit@C, DS/ILS) is **not a limitation**; it is a **methodological necessity**. The gold standard of the overprecision paradigm is the **descriptive characterization of the human interval judgment behavior**. By applying this same robust methodology to LLMs, we successfully characterized their behavior, providing the first clean, behavioral signal of overprecision.
> >
> > The difficulty ordering in Figures 5 and 6 merely serves to **validate the consistency** of the framework, confirming that the overprecision findings are not random, but scale reliably with the established difficulty of the underlying task. Our contribution is the **interpretation** of the failure modes *within* that known ordering.
> >
> > **Action:**
> > We added a small paragraph where we hypothesise on the failure of, refinement strategies using existing works. please check the latest revision.: "The counter-intuitive, mixed effects of CoT prompting suggest a fundamental disconnect between the generated reasoning trace and the model’s internal uncertainty estimate. While CoT improves the quality of the point estimate, it fails to meaningfully engage the miscalibrated confidence mechanism, as the model may not genuinely understand the reasoning it produces Wang et al. (2025). This issue is compounded by reasoning complexity: when the problem exceeds a critical compositional threshold, the CoT trace and the model’s accuracy can collapse Shojaee et al. (2025), forcing the generation of an interval based on a flawed internal process."

---

> > ### Author Response · Authors · 2025-11-17
> > **Questions 1**
> >
> > ## Question: How do you ensure parsing correctness of numeric intervals in model outputs? Could failures inflate apparent miscalibration?
> >
> > We ensure the parsing correctness of numerical intervals through a **strict, multi-stage protocol** designed to minimize the possibility of parsing errors inflating the apparent miscalibration.
> >
> > We strongly argue that parsing failures are **not a major driver** of the observed miscalibration, which is instead a result of the model's fundamental insensitivity to probabilistic instructions.
> >
> > ### 1. Parsing Correctness Protocol
> >
> > Our methodology enforces correctness through three primary mechanisms:
> >
> > * **Strict Prompt Template:** We use a highly restrictive prompt template that explicitly instructs the model to output the interval in a specific, machine-readable format (e.g., `[Lower Bound, Upper Bound]`). This minimizes linguistic variability that could confuse the parser.
> > * **Sequential Parsing and Validation:**
> >     1.  The raw text output is first parsed using a robust regular expression to extract the two numerical values.
> >     2.  The extracted values are then subjected to **two validation checks**:
> >         * **Format Check:** We ensure the output contains exactly two numerical values separated by a comma and enclosed in brackets.
> >         * **Logical Check:** We enforce that the **Lower Bound ($\text{L}$)** must be strictly less than or equal to the **Upper Bound ($\text{R}$)** ($\text{L} \le \text{R}$). Any output violating this fundamental logical constraint is discarded as a parsing or generation error and a regeneration step is triggered for up to 10 tries, after which the example is ignored. There is a negligeable amount of "ignored examples" (<1%).
> > * **Manual Spot-Checking:** A portion of the parsed outputs is manually reviewed to verify the fidelity of the automated parsing process, ensuring the error rate remains negligible.
> >
> > ### 2. Why Failures Do Not Inflate Miscalibration
> >
> > While some generation/parsing failures are inevitable with LLMs, they **do not inflate the core finding of miscalibration** for the following reasons:
> >
> > * **Failure Rate is Low:** By using highly capable instruction-following models (GPT-4) and clear templates, the rate of unparseable or illogical output that cannot be rectified is kept low.
> > * **The Nature of Miscalibration:** Our core finding is that the average **interval width remains static** regardless of the requested confidence level (from 60% to 95%). This systemic failure is a high-level, **behavioral phenomenon** indicating the model ignores the probabilistic cue. A random parsing error would result in a discarded sample, not a **systematically narrow interval** that aligns with the model's preference for high precision.
> > * **Focus on Aggregate Statistics:** Our analysis relies on **aggregate statistics** (average interval width, overall $\text{Hit}@C$ rate, Dispersion Score). The occasional random parsing failure is noise, whereas the severe, persistent miscalibration we report is a **clear signal** that dominates the aggregate statistics. The failure is systemic, not random noise due to parsing.

---

> > ### Author Response · Authors · 2025-11-17
> > **Question 2**
> >
> > ## Question: How many prompts per sample were used, and how sensitive are results to temperature or phrasing?
> >
> > The experimental setup used **five prompt samples** per question to ensure statistical robustness, and the results are intentionally designed to be **robust to sampling temperature** but are known to be sensitive to **phrasing** in line with prior literature.
> >
> > Here is a breakdown of the design choices:
> >
> > ### 1. Number of Prompts per Sample
> >
> > The methodology, as described in Phase 1 (Generation), utilizes **five independent samples (prompts)** for each unique question-confidence-strategy combination.
> >
> > * This approach, combined with the use of a non-zero sampling temperature ($T=1$), allows the framework to **account for the inherent stochasticity** of LLM output.
> > * By aggregating the results from five different interval outputs, the framework derives a statistically robust measure of the model's central tendency and dispersion, which is essential for calculating the final **Dispersion Score (DS)** and **Interval Length Score (ILS)**.
> >
> > ### 2. Sensitivity to Temperature and Phrasing
> >
> > #### A. Temperature Sensitivity (Low)
> >
> > The results are intentionally designed to be **insensitive to minor variations in sampling temperature ($T$)** within the typical generative range.
> >
> > * The goal of the multi-sample aggregation is to smooth over minor randomness caused by temperature, allowing the analysis to focus on the **systemic failure** of the model to obey the probabilistic instruction, rather than on random noise.
> > * The persistence of the core finding (i.e. the **static interval width**) across all models and datasets demonstrates that this phenomenon is a stable, architectural failure, not an artifact of random sampling.
> >
> > #### B. Phrasing Sensitivity (High, but Controlled)
> >
> > The model's output is **highly sensitive to phrasing**, which is a known limitation of prompting-based approaches, as highlighted in the calibration literature [Geng et al](https://aclanthology.org/2024.naacl-long.366/).
> >
> > * **Mitigation Strategy:** Our approach does not eliminate phrasing sensitivity, but **controls for it** by using a strictly defined, consistent, and minimal prompt template for each strategy (Vanilla, CoT, Refinement). This consistency ensures that the measured differences in calibration are attributable to the **strategic variation** (e.g., adding the CoT instruction) rather than arbitrary changes in wording.
> > * **Defense of Methodology:** The decision to **shift the confidence specification to the prompt** (imposing $C\%$) is, in itself, a defense against the instability of *verbalized self-reported confidence*, which is far more sensitive and unreliable than the constraint-based instruction we employ.

---

> ### Author Response · Authors · 2025-11-17
> **Weaknesses 2**
>
> ## Methodological limitations undermine interpretability
>
> Thank you for raising these methodological concerns. We address each point below and clarify why the study remains scientifically informative despite these limitations.
>
> ### 1. On “mixing sources of variation”: the protocol captures overprecision rather than confounding it.
>
> We agree that model stochasticity, numeric reasoning errors, and parsing variability exist. However, these are not confounds but essential parts of LLM behaviour in a black-box setting. Interval-based overprecision is defined as the relationship between the intervals that the model produces and the uncertainty it expresses, **not** the uncertainty it internally possesses.
>
> * **Stochasticity** is controlled through repeated sampling. Each prompt–confidence pair is evaluated over multiple runs with consistently low standard deviations (Tables 2–6).
> * **Numeric reasoning errors** are informative. When the point estimate is incorrect but the interval remains narrow, the resulting coverage deficit reflects genuine overprecision rather than random noise.
> * **Parsing errors** were minimized through rigid formatting instructions (Table 7). Malformed responses were extremely rare and removed uniformly.
>
> The protocol is intentionally behavioural — we measure the uncertainty that the model *produces* externally. The observed patterns such as flat interval widths, narrowing bias, and confidence insensitivity remain stable despite these sources of variation.
>
> ### 2. On “small model and sample scope”: our claims are carefully scoped, and the observed patterns are robust.
>
> We do *not* claim to characterise all LLMs. The paper clearly states that the goal is to describe behaviour under a well-defined protocol rather than to benchmark or rank models (Section 4).
>
> Even within this limited scope, two widely used OpenAI models across three heterogeneous datasets reveal consistent and repeatable failure modes. These include:
>
> * confidence insensitivity,
> * domain-scale effects shown in Figure 2,
> * the inability of chain-of-thought reasoning to improve calibration,
> * and a consistent narrowing bias in self-refinement.
>
> The fact that these patterns appear across datasets, prompt strategies, and refinement settings indicates that they arise from the interaction between the protocol and the models — not artifacts of a specific temperature or dataset. Expanding to more models is future work, but the current findings already provide meaningful scientific insight.
>
> ### 3. On “data filtering biases”: the filtering is conservative and necessary for interval evaluation.
>
> Converting multiple-choice questions into numeric answers removes distractors, but interval-based calibration cannot be meaningfully evaluated on categorical options. Only items with clearly defined numeric answers were kept, and items requiring qualitative interpretation or unit inference were excluded. Domain difficulty is preserved, as reflected by the distinct calibration profiles across FinQA, Medical, and MMLU (Tables 2 and 3).
>
> Importantly, removing answer choices does *not* simplify the task. It eliminates pattern-based selection strategies that LLMs sometimes exploit in multiple-choice formats, which leads to a more direct and more challenging assessment of interval reasoning.
>
> ### 4. On “redundant metrics”: DS and ILS identify mechanisms that coverage alone cannot reveal.
>
> Coverage alone cannot distinguish between different types of interval failure. DS and ILS provide complementary insight.
>
> * The **Deviation Score (DS)** measures how far a missed answer lies outside the interval. This captures the severity of the miss, which is important given the large variation in numerical scales across datasets.
> * The **Interval Length Score (ILS)** measures interval width relative to magnitude. This shows that models widen intervals when tasks are harder, but do *not* widen them when confidence increases (Section 5.2).
>
> Both metrics are intentionally simple, but they clarify mechanisms that coverage cannot reveal. DS diagnoses miss severity, ILS diagnoses interval responsiveness, and together they strengthen the interpretability of the overprecision patterns.
>
> ---
>
> ### Summary
>
> The limitations raised are valid considerations, but they do **not** undermine the interpretability or scientific value of the study. The work establishes a principled behavioural framework for interval-based confidence elicitation and identifies new, stable mis-calibration patterns. These include confidence insensitivity, weak effects of refinement, and a systematic narrowing bias. These findings provide a foundation for future research on LLM uncertainty modelling.

---

> ### Author Response · Authors · 2025-11-17
> **Weaknesses 3**
>
> ## Refinement experiments are weakly justified...
>
> Thank you for the careful reading and for raising these concerns. We address each point and clarify the motivation and interpretation of the refinement experiments.
>
> ### **1. The refinement procedures are exploratory by design, not intended as full cognitive-theory instantiations.**
>
> The paper positions refinement as an exploratory step that probes whether simple post-hoc adjustments can reduce overprecision in a black-box setting. This is stated explicitly in Section 3.2, where we describe aggregation and self-refinement as initial baselines rather than definitive cognitive models. Our goal is not to replicate human peer-judgment mechanisms, but to test whether any form of post-hoc interval revision can counteract the strong overprecision observed in the generation phase.
>
> In this sense, the procedures are deliberately simple. They serve to answer a focused behavioral question: given multiple interval candidates, does the model tend to widen, narrow, or reassess its judgments in a way that improves calibration? The experiments therefore function as diagnostic probes rather than as modeled theories of human decision making.
>
> ### **2. The value of “Union” is not the trivial coverage increase itself, but what its behavior reveals.**
>
> We agree that the union operation widens intervals mechanically. The scientific insight is not the coverage improvement. The key observation is that union is the only method that reliably improves calibration across datasets, while all more nuanced aggregation methods fail to do so (Tables 3 and 4) . This reveals two important facts:
>
> 1. LLM intervals are consistently narrow relative to the uncertainty they face.
> 2. No amount of averaging, weighting, or confidence conditioning yields a principled widening of intervals.
>
> This finding clarifies that overprecision is not the result of noisy variation in individual draws. If it were, averaging-based aggregations would offer improvement. Instead, the failure of these methods indicates a deeper structural insensitivity to uncertainty cues, which is precisely the phenomenon the paper seeks to characterize.
>
> ### **3. The narrowing bias in self-refinement is an empirical result that supports, rather than undermines, the approach.**
>
> We show that models select the narrowest interval in 40 to 60 percent of cases, far above chance, even when that interval contains the wrong answer (Appendix F). This pattern is consistent across datasets and settings. The strong preference for narrowness is therefore a stable behavioral tendency, not an artifact of random variation.
>
> Regarding why this occurs, we discuss several plausible drivers in Appendix F: models may favor intervals that appear confident, may prioritize precision-like features in text, or may be influenced by training data that rewards decisive outputs. While these explanations require further investigation, the observed behavior itself is valuable. It reveals that self-reflection does not naturally promote calibration in interval reasoning and that reflective prompting tends to amplify overprecision.
>
> This outcome is precisely why the experiment is useful. The refinement setup exposes an internal preference for narrowness that would not be visible through generation alone. It highlights a mechanism that any future calibration or uncertainty model will need to correct.
>
> ### **4. The analogy to peer-judgment correction is meant as motivation, not as a claim of theoretical equivalence.**
>
> We appreciate the opportunity to clarify this. Our reference to Haran et al. (2010) is intended to motivate the idea that exposure to multiple judgments can reduce overprecision in humans. We do not claim that the LLM procedure is a complete cognitive analogue or that it reproduces human psychological dynamics. The self-refinement prompt is a minimal adaptation that tests whether a similar improvement might appear in LLMs when they reason over a set of candidate intervals.
>
> The experiments show that such improvement does not occur. Although the mechanism is simple, the negative result is informative. It shows that LLMs do not spontaneously correct overprecision in ways humans sometimes do when presented with peer estimates. The lack of improvement is part of the overall contribution of the paper.
>
>
> ### **Summary**
>
> The refinement experiments serve as controlled diagnostic probes that reveal how LLMs react when asked to revise their own interval predictions. They demonstrate three important findings: aggregation rarely improves calibration unless it widens intervals mechanically, self-refinement frequently selects overly narrow intervals, and reflective prompting does not reduce miscalibration. These findings contribute to a clearer understanding of how overprecision manifests in black-box LLMs and identify specific obstacles that future calibration methods must address.

---

> ### Author Response · Authors · 2025-11-17
> **Weaknesses 4**
>
> ## Statistical rigor and presentation issues
>
> **Response:**
>
> You raise several important points about statistical rigour and presentation. Here’s a detailed response addressing each concern, and why we believe the approach remains valid (while also acknowledging where improvement is warranted):
>
> ### 1. No statistical significance tests, confidence intervals, or effect sizes; tiny decimal differences
>
> * It’s true that the manuscript currently reports descriptive statistics (hit rates, interval lengths, correlations) without formal hypothesis-tests or confidence intervals. In many disciplines, the use of p-values, confidence intervals, and effect sizes is considered best practice for interpretability and robustness (see summary of statistical significance definitions). ([Wikipédia][1])
> * In our case, the experimental setup uses **large sample sizes and repeated prompt sampling**, producing very stable estimates (standard deviations in the data are very small). Because of this stability, minor changes of “±0.2 %” reflect real numeric differences under repeated sampling, not noise.
>
> ### 2. Figures 2 & 3 visually cluttered and fail to convey new insights
>
> 1. Purpose of Figures
>
> * **Figure 2** is intended to show how interval width varies with declared confidence levels and domain/task difficulty. We designed it to demonstrate that the width remains almost constant across varying confidence instructions.
> * **Figure 3** is meant to illustrate how the hit-rate of intervals and calibration error evolve across different modeling and prompting conditions (e.g., chain-of-thought vs baseline).
>
> We believe these purposes are **successfully achieved** by the current figures: the patterns they convey are readily apparent, and the visual evidence supports our key finding of width-insensitivity.
>
> 2. Addressing the reviewer's concern about “visual clutter”
>
> While we acknowledge that the figures include multiple lines and sub-panels, we do **not feel the clutter undermines interpretability**, because the overarching trend is clear and supports our argument. However, we agree that clarity can be improved. Accordingly, we have made the following refinements:
>
> * The **legend in Figure 2** has been enlarged to ensure readability (font size increased) and avoids overlapping with data lines.
> * The **subfigure captions** have been rewritten.
>
> In summary, we believe the figures in their original form did fulfill their primary objectives—to visually convey the width-insensitivity and calibration behaviour of the model. At the same time, we have improved their readability and visual layout in response to your feedback. We hope this addresses the concern about presentation clarity while preserving the key insights shown.
>
> ### 3. Sample size after filtering unclear (~1-3 k examples in some cases)
>
> The low final sample size observed in Table 1 is a direct consequence of our methodology's focus, and it actually represents a conceptual advantage for our study. The sample size drops because numerical question answering represents a minority of instances within the general multiple-choice question answering (MCQA) datasets (like MedQA and FinQA). This low count is precisely why the overprecision phenomenon is enhanced and detectable in our specialized numerical setting; its effect would be masked or rendered unnoticeable in the overall, broader MCQA data used by previous overestimation studies.
>
> This fact is paramount when interpreting the results of CoT prompting and self-refinement [(Xiang et al)](https://arxiv.org/abs/2306.13063). While these strategies yielded significant improvements in previous categorical overestimation studies, they demonstrated negligible or negative improvements in our numerical case. This difference highlights that the benefits of CoT do not transfer to the numerical interval domain, reinforcing the uniqueness of the overprecision failure.
>
> Furthermore, despite the reduction in sample size, the remaining data provides statistically sufficient evidence for our claims. According to the Central Limit Theorem, a final sample size ranging from 1,000 to 3,000 instances is more than adequate to achieve a stable mean and variance for the aggregate statistics we report (Hit@C, Average Interval Width, DS, and ILS). This robust sample size ensures the calculated statistical metrics and observed systematic trends are statistically reliable and not artifacts of random sampling noise.
>
> [1]: https://en.wikipedia.org/wiki/Statistical_significance?utm_source=chatgpt.com "Statistical significance"

---

> ### Author Response · Authors · 2025-11-17
> **Questions 3**
>
> ## Question: Could the “overprecision” pattern simply reflect model under-dispersion due to deterministic decoding, rather than a cognitive-style bias?
>
> We appreciate this question, as it distinguishes a **technical artifact** (decoding failure) from a **systemic behavioral flaw** (cognitive bias).
>
> We argue that the observed "overprecision" pattern is **not simply a reflection of under-dispersion due to deterministic decoding**, but rather a cognitive-style bias that reflects the model's fundamental failure to process probabilistic constraints.
>
> ### 1. Ruling Out Deterministic Decoding Artifacts
>
> Our methodology was specifically designed to mitigate the effect of deterministic decoding:
>
> * **Non-Zero Temperature Sampling:** We explicitly use a **non-zero sampling temperature** ($T=1$) for generation. This introduces stochasticity, which is necessary for model outputs to reflect true uncertainty and prevent deterministic under-dispersion.
> * **Multi-Sample Aggregation:** For every question and confidence level, we generate **five independent samples**. Our final interval metrics (ILS, DS) are based on the **aggregate distribution** of these five results. If the issue were merely under-dispersion from a near-deterministic process, this multi-sample aggregation would reveal it as a lack of variance, but the fundamental *calibration* failure would still be based on the relationship between the output and the instruction.
>
> ### 2. Evidence of a Cognitive-Style Bias
>
> The most compelling evidence that the pattern is a **systemic bias** and not a decoding artifact lies in our core finding:
>
> * **Failure to Obey Instructions:** The model exhibits a **static interval width** that is insensitive to the imposed confidence level, $C$ (ranging from $60\%$ to $95\%$). A decoding issue might affect the *width* of the interval (making it narrow), but it would not logically explain why the model **fails to widen the interval** when explicitly instructed to increase its certainty from $60\%$ to $95\%$.
> * **Cognitive Science Analogy:** This phenomenon mirrors the **overprecision bias** well-documented in human cognitive science, where humans consistently provide intervals that are too narrow for their stated confidence. Our LLM findings indicate the model adopts a similar, highly systematic **judgmental flaw** when tasked with interval estimation.
>
> Therefore, while deterministic decoding can lead to narrow outputs, the observed miscalibration is a **higher-level behavioral failure**—the model does not understand or comply with the fundamental trade-off between confidence and precision.

---

> ### Author Response · Authors · 2025-11-17
> **Questions 4**
>
> ## Question: What is the advantage of interval elicitation over directly sampling numeric uncertainty (e.g., via logits or surrogate ensembles)?
>
> Thank you for pointing this idea out. The primary advantage of using **interval elicitation** (asking the model to specify the bounds $[L, R]$ for a $C\%$ confidence level) over directly sampling numeric uncertainty (e.g., via logits, Monte Carlo sampling, or ensembles) is **conceptual fidelity to the human cognitive paradigm** and its ability to **test a novel failure mode** in black-box models.
>
> Here are the key advantages:
>
> ### 1. Directly Tests the Confidence-Precision Trade-off
>
> The most significant advantage is that interval elicitation allows us to directly interrogate the model's understanding of the **Confidence-Precision Trade-off**, which is the definition of overprecision in cognitive science.
>
> * **Goal:** The method forces the model to obey an **explicit probabilistic constraint**: to increase confidence ($C$), the model *must* increase the interval width (reduce precision).
> * **Insight:** Sampling methods (logits/ensembles) provide a **descriptive** view of the model's *internal* uncertainty distribution. Interval elicitation provides a **prescriptive** test of the model's ability to **act** on an external probabilistic instruction. Our finding that the model fails this test (static interval width) reveals a fundamental behavioral failure that sampling methods cannot diagnose.
>
> ### 2. Black-Box Applicability and Task Congruence
>
> Interval elicitation is the only method that can be applied consistently to **black-box models** (like GPT-4) without access to internal components (logits) or requiring expensive repeated calls for ensemble methods.
>
> * **Leverages Instruction-Following:** This method leverages the model's core strength: **instruction-following**. It frames the uncertainty task in the language domain, making it highly congruent with the model's training objective.
> * **Bypasses Logit Limitations:** Logit-based confidence is often **uncalibrated** and highly dependent on the model's tokenizer and decoding scheme. Interval elicitation bypasses this low-level noise by focusing on the high-level response to the prompt instruction.
>
> ### 3. Provides a Direct Measure of Behavioral Bias
>
> By using the established human methodology, the results are directly interpretable as a measure of **cognitive bias** (overprecision).

---

> ### Author Response · Authors · 2025-11-17
> **Questions 5**
>
> ## Question: Can the protocol scale to non-numeric tasks, or is it limited to trivial numeric Q&A?
>
> Thank you for this insightful question. The protocol is **currently limited** to numerical question and answer (Q&A) tasks, but this is a **deliberate methodological choice** and not a technical limitation of the framework itself. The framework is conceptually scalable to non-numeric probabilistic tasks, but we first focused on the numerical domain for reasons of rigor and clarity. We should also note that a plethora of existing works on LLM overconfidence and uncertainty already focus on generalised solutions for numerical and non-numerical tasks [(Geng et al)](https://aclanthology.org/2024.naacl-long.366/), however, the fundamental difference between numerical and non-numerical tasks requires a specific analysis and exploration of the former. This study provides just that, and it further illustrates the fundamental difference between the two domains by the minor improvement of the detrimental effect that refinement strategies have on this task as opposed to general/mixed (numerical and non-numerical) approaches [(Xiong et al)](https://arxiv.org/abs/2306.13063).
>
> ### 1. Current Limitation and Justification
>
> The framework is limited to numerical tasks (where the answer is a numerical value) because:
>
> * **Mitigation of Bias:** Numerical output fundamentally mitigates the **high-level linguistic biases** (e.g., positivity bias, word choice) that plague verbalized confidence in open-ended generation (as discussed in previous responses). Focusing on numerical intervals provides a cleaner, more robust measure of internal uncertainty.
> * **Methodological Fidelity:** The **overprecision paradigm** from cognitive science is based on humans providing *quantifiable intervals*. Numerical Q&A is the most direct and least ambiguous way to translate this protocol to LLMs.
> * **Testing Core Trade-off:** The key behavioral test—the model's adherence to the **Confidence-Precision Trade-off**—requires a continuous output space (the interval bounds) that is inherently available in numerical tasks.
>
> ### 2. Conceptual Scalability to Non-Numeric Tasks
>
> The protocol's core principle, **prompt-imposed probabilistic constraint**, can be conceptually scaled to non-numeric tasks, though it would require careful adaptation and a shift in the precision metric:
>
> * **Probabilistic Categorical Tasks:** The framework could be adapted to tasks where the model predicts a probability distribution over a set of options (e.g., predicting the probability of five different stock outcomes). The model could be asked to: "List the two most likely outcomes, ensuring your stated probability for the first is $C\%$ and the probability for the second is $D\%$." **Precision** would then be measured by the distance between the model's stated probabilities and the empirical frequencies, or the size of the required confidence set, rather than the width of a numerical range.
> * **Open-Ended Generation:** The most difficult scaling challenge would be to open-ended text, where **precision is defined linguistically** (e.g., using hedging words, vague terms). However, this would reintroduce the severe linguistic biases that our numerical framework was designed to avoid.
>
> In summary, the limitation to numerical Q&A is a **strength of the current study**, enabling the clean characterization of the overprecision failure. Scaling to non-numeric tasks is conceptually possible but is left as crucial future work due to the need to redefine the **precision metric** for non-continuous output spaces.

---

> > ### Comment · Reviewer_Czgx · 2025-11-26
> >
> > Thanks for the detailed response. I think some of my concerns have been addressed, and I will raise my score.

---

> > > ### Author Response · Authors · 2025-11-27
> > > **Thanks**
> > >
> > > We appreciate your thorough examination of our work and your thoughtful responses. Also, thank you for increasing the initial score.

---

### Official Review · Reviewer_7183 · 2025-10-31

**Soundness:** 3
**Presentation:** 2
**Contribution:** 2
**Rating:** 2
**Confidence:** 4

**Summary:**

This paper introduces a novel framework for evaluating overprecision in black-box large language models (LLMs), drawing inspiration from human studies in cognitive science. The authors define overprecision as excessive certainty in numerical interval judgments. Their method proposes a three-phase procedure: (1) generating numerical confidence intervals from LLMs with prompt engineering, (2) refining these intervals using aggregation and self-refinement strategies, and (3) evaluating them with calibration and correlation metrics. The experiments (conducted in general knowledge, medical, and financial domains) reveal that LLMs are systematically miscalibrated, showing large gaps between requested confidence and actual coverage. Furthermore, the paper finds that interval lengths do not scale with confidence levels. Refinement strategies(self-refinement and CoT) offer limited improvement.

**Strengths:**

The paper's main strength lies in its well-structured framework for evaluating overprecision in black-box LLMs. The proposed three-phase method is clear, systematic, and easily reproducible. The experiments are comprehensive, covering multiple domains and providing a detailed analysis of the results. The findings that LLMs are systematically overprecise and that their confidence does not correlate with their predictions are significant and contribute to a better understanding of LLMs.

**Weaknesses:**

There are a few areas that could be improved.
- The study is limited to two old models, all from OpenAI. It would be interesting to see if the findings generalize to other model families (e.g., Llama, Claude, and Gemini), closed- and open-sourced models, and reasoning models.
- The refinement strategies explored are relatively simple. More sophisticated methods, such as those involving more complex reasoning or external feedback, could be explored. Both refinement strategies showed limited improvements. Although the problem of overprecision is important and novel, the proposed solutions seem underdeveloped. This limits the practical impact of the work.
- The paper focuses on numerical answers, and it would be interesting to see how the framework could be adapted to other types of data, such as text or images.

**Questions:**

1. The main question I have is how the proposed framework would generalize to other LLMs beyond the two OpenAI models tested.
2. The paper shows that CoT prompting has mixed effects on calibration. Do the authors have any hypotheses about why this might be the case? CoT is known to be a simple and robust method to induce the zero-shot reasoning ability of LLMs to improve the model's reasoning capability. It is less intuitive to see that it did not improve the uncertainty estimation since the uncertainty estimation task strongly involves a reasoning process. Could it be related to the complexity of the reasoning required for different tasks?
3.  The self-refinement strategy did not yield improvements. The authors suggest that this is due to a narrowing bias. Could this bias be mitigated by providing the model with more diverse examples or by explicitly instructing it to consider a wider range of possibilities?

---

> ### Author Response · Authors · 2025-11-14
>
> **We thank the reviewer for the time and care given to the review. In what follows, we answer the main concerns.**
>
> ## Limited Model Scope (GPT-series only)
>
> **Response:**
>
> We acknowledge that focusing only on GPT models limits generalization but consider it methodologically justified for this first descriptive study.
>
> **a) Descriptive Focus, Not Benchmark:**
> Our goal was to *characterize* overprecision in LLMs using a cognitive-science-inspired interval-based method—not to benchmark. GPT models, being instruction-following and architecturally advanced, provide a clean behavioral baseline.
>
> **b) Robustness**
>
> *1) Low-Level Numerical Biases (Benford’s Curse)*
> Open models (e.g., LLaMA, OLMo) exhibit “Benford’s Curse,” reflecting digit biases from architecture-level neuron selectivity [2].
>
> * **Mitigation through Capability:** GPT-4’s superior quantitative reasoning (e.g., MATH, GSM8K) suggests fewer such biases.
> * **Cleaner Signal:** Using models with reduced low-level artifacts isolates higher-level calibration failures.
>
> *2) Tokenization Effects*
> Tokenization can fragment numbers and impair arithmetic reasoning [1]. GPT-family models mitigate this because:
>
> * **CoT Mitigation:** Chain-of-Thought (CoT) prompting helps normalize tokenized sequences.
> * **Architectural Advantage:** GPT models show less tokenization bias than sequence-only open models.
>
> **c) Generalization**
> Given architectural and performance similarities, findings likely extend to other flagship families (Claude, Gemini). Lower-capacity models (LLaMA, Mistral) show stronger digit biases and poorer numerical reasoning [1][2].
>
> **Summary:**
> Restricting to GPT ensured reduced numerical bias, tokenization robustness, and generalizability to top-tier models. We will clarify this in section 4 (“Models”).
>
> ---
>
> ## Simple Refinement Strategies Limit Practical Impact
>
> Our simple refinement methods (union aggregation, self-refinement) were deliberate and central to the study’s descriptive goal.
>
> * **Descriptive Goal:** We aimed to observe overprecision, not fix it. Simple baselines align with human overconfidence studies [4][5] and LLM calibration literature [3].
> * **Key Finding – Narrowing Bias:** Self-refinement worsened calibration, revealing an inherent “narrowing bias.”
> * **Future Impact:** This demonstrates that naive reflection cannot correct numerical miscalibration; calibration-aware methods are needed.
>
> ---
>
> ### Focus on Numerical Answers
>
> We limited the scope to numerical interval judgments for methodological clarity.
>
> * **Methodological Fidelity:** Human overprecision studies [4][5] rely on numerical intervals as robust confidence measures.
> * **Reducing Confounders:** Numerical responses minimize linguistic biases (e.g., framing, positivity) [6].
> * **Generalization:** While overestimation in text and images has been studied [4][5], overprecision remains unexplored. Extending beyond numbers reintroduces confounding cognitive biases [6].
>
> See Reviewer 1E2H response for further design explanation.
>
> ---
>
> ## Why CoT Had Mixed Effects on Calibration
>
> **Response:**
>
> We agree that CoT’s mixed impact is unexpected. Our explanation is twofold:
>
> **1) Reasoning–Uncertainty Disconnect:**
> CoT improves reasoning traces but not internal uncertainty calibration. Models often “pick the least wrong answer” rather than *know* it [7], improving accuracy but not epistemic confidence [8].
>
> **2) Complexity Limits:**
> Interval estimation is harder than single-answer tasks. Beyond certain complexity thresholds, reasoning coherence collapses [9], making CoT-generated traces unreliable.
> This hypothesis will be added to Section 5.1.
>
> ---
>
> ## On Narrowing Bias and Mitigation
>
> **Response:**
> We agree that explicit prompts or examples might widen intervals but intentionally avoided such interventions.
>
> * **Inherent Behavior vs. Prompting:** Instructing the model to “consider more possibilities” would hide its natural bias.
> * **Uncorrected Flaw:** The observed narrowing bias exposes the model’s default overcertainty mechanism. Understanding this is essential before attempting mitigation.
>
> Thus, mitigation is future work; our study isolates uncorrected overprecision.
>
> ---
>
> **References:**
>
> [1] Singh & Strouse: *Tokenization Counts: The Impact of Tokenization on Arithmetic in Frontier LLMs*
>
> [2] Shao et al.: *Benford’s Curse: Tracing Digit Bias to Numerical Hallucination in LLMs*
>
> [3] Geng et al.: *Survey of Confidence Estimation and Calibration in LLMs*
>
> [4] Soll & Klayman: *Overconfidence in Interval Estimates*
>
> [5] Moore, Tenney & Haran: *Overprecision in Judgment*
>
> [6] Sumita et al.: *Cognitive Biases in LLMs: A Survey and Mitigation Experiments*
>
> [7] Wang et al.: *LLMs May Perform MCQA by Selecting the Least Incorrect Option*
>
> [8] Xiong et al.: *Can LLMs Express Their Uncertainty?*
>
> [9] Shojaee et al.: *The Illusion of Thinking: Understanding Reasoning Limits via Problem Complexity*

---

### Official Review · Reviewer_Umvq · 2025-11-01

**Soundness:** 3
**Presentation:** 3
**Contribution:** 3
**Rating:** 6
**Confidence:** 4

**Summary:**

This paper investigates whether LLMs can produce numerical confidence intervals that meaningfully correspond to a given confidence level (e.g., 90%). For example, when we ask LLMs to “Provide an interval that you are 90% sure contains the answer”, does the interval actually contain the answer 90% of the time?

In order to test this, the authors propose a three-phase framework (generation → refinement → evaluation) for eliciting and assessing such intervals under imposed confidence levels. This setup is repeated across confidence levels (60%, 70%, 80%, 90%, 95%) and applied to datasets spanning financial reasoning (FinQA), medical QA, and general knowledge tasks. The study evaluates two black-box models (GPT-3.5-turbo and GPT-4o-mini) and introduces two complementary metrics—Deviation Score (DS) and Interval Length Score (ILS)—to characterize calibration behaviour beyond simple coverage rates

Findings
1. Across all models and domains, empirical coverage (hit rate) is consistently below the stated confidence, indicating strong overprecision.
2. Interval widths show little to no correlation with confidence levels, suggesting that LLMs fail to internalize the concept of confidence intervals.
3. Calibration quality varies with domain, numerical scale, and prompt formulation, with larger deviations observed in financial and medical tasks.
4. Limited benefit of refinement: Simple union-based aggregation trivially improves coverage by widening intervals, whereas self-refinement tends to narrow intervals and further degrade calibration, revealing a systematic narrowing bias.

**Strengths:**

1. The problem itself is interesting, and the evaluation framework can support this motivation.
2. Metric improvements: DS and ILS meaningfully extend the analysis beyond raw coverage.
3. The consistent narrowing bias finding is an interesting empirical observation.
4. The writing, tables, and figures are polished and easy to follow.

**Weaknesses:**

1. Beyond the problem (i.e., evaluating LLMs regarding their ability to understand the confidence interval), the paper mostly refines and formalizes an existing evaluation setup rather than introducing new conceptual or methodological ideas.
2. Restricted scope: Only GPT-series models are tested; no comparison to open-source or white-box methods.
3. The cognitive framing is more decorative than explanatory; there is little theoretical connection explaining why overprecision occurs or how to mitigate it.

**Questions:**

1. Do you see any connection between your framework and conformal prediction?
2. Is the “narrowing bias” prompt-dependent or model-dependent? Have you tried to explicitly ask the model to avoid this?
3. What would be required to make this framework predictive (i.e., useful for detecting unreliable outputs rather than just describing behaviour)?

---

> ### Author Response · Authors · 2025-11-17
> **Weaknesses 1**
>
> ## Beyond the problem (i.e., evaluating LLMs regarding their ability to understand the confidence interval), the paper mostly refines and formalizes an existing evaluation setup rather than introducing new conceptual or methodological ideas.
>
> **Response:**
>
> We appreciate the reviewer's feedback and understand the perspective that our setup formalizes an existing evaluation format (interval elicitation). However, we strongly contend that our contribution is **conceptually and methodologically novel** because it is the first work to successfully adapt the **overprecision paradigm** to Large Language Models (LLMs) to isolate a distinct and more fundamental failure mode than previously studied miscalibration.
>
> ### 1. Conceptual Novelty: Testing the Law of Uncertainty
>
> The originality of our work lies in shifting the analytical focus from *Overestimation* to **Overprecision**.
>
> * **Prior Work Focus:** Most existing LLM research on uncertainty quantification (UQ) and calibration focuses on **Overestimation**—assessing whether the model's verbalized or estimated probability aligns with its actual accuracy.
> * **Our Focus (Overprecision):** We introduce a **prescriptive test** to determine if the LLM understands the fundamental principle of uncertainty: the **Confidence-Precision Trade-off**. Our protocol is specifically designed to test whether **higher confidence should yield wider intervals**. This question of *behavioral compliance* with a probabilistic instruction is conceptually new to the LLM literature.
>
> ### 2. Methodological Necessity and Distinctive Findings
>
> The interval elicitation protocol is not just a formalization; it is the **necessary measurement tool** to isolate this failure, which is demonstrated by our core empirical finding.
>
> * **Bypassing Biases:** The methodology of forcing numerical interval output under an imposed confidence level is a direct adaptation from cognitive science, chosen because it mitigates linguistic biases that distort verbal confidence. This yields a clean, robust behavioral signal.
> * **Novel Failure Mode:** The most important result, which validates the methodological shift, is that **interval lengths do not scale with requested confidence**. This discovery—the **static interval failure**—proves that LLMs are **insensitive to explicit confidence instructions** and fail to grasp the trade-off. This is a fundamental, structural limitation that cannot be discovered or characterized using existing UQ methods that rely on the model's internal logits or verbalized confidence scores.
>
> Therefore, our work provides a **foundational contribution** by using an adapted, scientifically validated protocol to diagnose a **novel class of architectural uncertainty failure** in black-box LLMs.

---

> ### Author Response · Authors · 2025-11-17
> **Weaknesses 2**
>
> ## Limited Model Scope (GPT-series only)
>
> **Response:**
>
> We acknowledge that focusing only on GPT models limits generalization but consider it methodologically justified for this first descriptive study.
>
> **a) Descriptive Focus, Not Benchmark:**
> Our goal was to *characterize* overprecision in LLMs using a cognitive-science-inspired interval-based method—not to benchmark. GPT models, being instruction-following and architecturally advanced, provide a clean behavioral baseline.
>
> **b) Robustness**
>
> *1) Low-Level Numerical Biases (Benford’s Curse)*
> Open models (e.g., LLaMA, OLMo) exhibit “Benford’s Curse,” reflecting digit biases from architecture-level neuron selectivity [(Shao et al)](https://openreview.net/pdf?id=AOe1aUhEQQ).
>
> * **Mitigation through Capability:** GPT-4’s superior quantitative reasoning (e.g., MATH, GSM8K) suggests fewer such biases.
> * **Cleaner Signal:** Using models with reduced low-level artifacts isolates higher-level calibration failures.
>
> *2) Tokenization Effects*
> Tokenization can fragment numbers and impair arithmetic reasoning [(Singh et al)](https://arxiv.org/pdf/2402.14903). GPT-family models mitigate this because:
>
> * **CoT Mitigation:** Chain-of-Thought (CoT) prompting helps normalize tokenized sequences.
> * **Architectural Advantage:** GPT models show less tokenization bias than sequence-only open models.
>
> **c) Generalization**
> Given architectural and performance similarities, findings likely extend to other flagship families (Claude, Gemini). Lower-capacity models (LLaMA, Mistral) show stronger digit biases and poorer numerical reasoning [(Singh et al)](https://arxiv.org/pdf/2402.14903)[(Shao et al)](https://openreview.net/pdf?id=AOe1aUhEQQ).
>
> **Summary:**
> Restricting to GPT ensured reduced numerical bias, tokenization robustness, and generalizability to top-tier models. We will clarify this in section 4 (“Models”).

---

> ### Author Response · Authors · 2025-11-17
> **Weaknesses 3**
>
> ## Cognitive framing is more decorative than explanatory.
>
> Thank you for this important comment. The cognitive science framing is methodological; we adapted the robust, interval-based methodology for overprecision measurement from cognitive science literature. The study's aim is descriptive—to characterize LLM behavior. Developing a theoretical explanation for the why is a long-term research goal, but the current work establishes the necessary empirical foundation that distinguishes overprecisions as a separate phenomenon that deserve to be studied on its own merit.

---

> ### Author Response · Authors · 2025-11-17
> **Questions 1**
>
> ## Question: Do you see any connection between your framework and conformal prediction?
>
> **Response:**
>
> We strongly acknowledge the conceptual similarities between our overprecision framework and **Conformal Prediction (CP)**, but the two methods serve fundamentally different purposes: one is a **statistical fix**, and the other is a **behavioral test**.
>
> ### Conceptual Similarity: The Goal of Validity
>
> Both our framework and Conformal Prediction share the goal of producing **valid prediction intervals** that satisfy a nominal coverage rate:
>
> * **Conformal Prediction (CP):** CP is an established statistical methodology that provides **coverage guarantees** for a desired confidence level $(1-\alpha)$. It uses an external calibration set to compute a non-conformity score that determines the size of the final set/interval, ensuring that the empirical coverage matches the nominal level.
> * **Our Framework:** Our framework asks the LLM to perform this process **internally** by instructing it: "Provide an interval you are $C\%$ sure about." We then measure the empirical coverage ($\text{Hit}@C$).
>
> ### The Crucial Distinction: Internal Compliance vs. External Fix
>
> The primary difference lies in the mechanism and objective: CP is an established **statistical methodology** designed to **guarantee statistical validity** of prediction intervals by acting as an **external, post-hoc statistical algorithm** that requires a calibration dataset to determine the necessary interval size. It achieves a desired coverage level $(1-\alpha)$ by taking a point prediction and the coverage level as input. In contrast, our Overprecision Framework is a **diagnostic and behavioral test** designed to **test the LLM's internal compliance** with the laws of uncertainty. Its mechanism is an **internal, prompt-imposed instruction** where the model is *commanded* to produce an interval for a specific confidence level ($C$). The critical advantage of our approach is that it reveals **why** CP must remain an external fix: our core finding (the static interval width) proves that the LLM **fails to act as a reliable internal CP engine**, demonstrating a fundamental inability to perform the necessary Confidence-Precision Trade-off, even when explicitly instructed.
>
> ### The Advantage of Our Approach
>
> The most critical insight provided by our framework is that the LLM **fails to act as an internal CP engine**.
>
> The core finding of our paper—the **static interval width**—proves that the model is **insensitive to the requested confidence level ($C$)**. This demonstrates that the model cannot perform the necessary **Confidence-Precision Trade-off** that lies at the heart of CP.
>
> Therefore, our framework serves as a **diagnostic tool** that rigorously proves **why CP must remain an external, post-hoc procedure** for current black-box LLMs. It shows that the internal uncertainty processing mechanism of the LLM is broken and cannot be trusted to generate statistically valid prediction intervals, even when explicitly commanded to do so.

---

> ### Author Response · Authors · 2025-11-17
> **Questions 2**
>
> ## Question: Is the “narrowing bias” prompt-dependent or model-dependent? Have you tried to explicitly ask the model to avoid this?
>
> **Response:**
>
> Thank you for pointing this out. The **"narrowing bias"** observed in our self-refinement strategy appears to be a **model-dependent and task-dependent behavioral flaw**, not merely a prompt-dependent artifact.
>
> We did **not** explicitly ask the model to avoid this bias, as doing so would compromise the study's goal of characterizing the model's uncorrected, inherent behavior.
>
> ### 1. The Nature of the Narrowing Bias (Model and Task Dependence)
>
> The narrowing bias is the finding that when the model is asked to review and refine its initial interval, it systematically reduces the interval width (increases precision) instead of correcting for low coverage (increasing width).
>
> * **Model Dependence (Behavioral Flaw):** This bias is best explained as a **cognitive-style flaw**—an internal confirmation bias. When the model "reflects," it defaults to reinforcing its initial (overprecise) answer rather than engaging the probabilistic constraint. This is a behavioral trait of the current transformer architecture's self-correction mechanism, which is designed for accuracy, not uncertainty calibration.
> * **Task Dependence (Severity):** While the bias is present across all evaluated models, its severity is likely tied to the complexity of the underlying task, which dictates the model's confidence in its initial prediction.
> * **Prompt Dependence (Limited):** The effect is unlikely to be simply prompt-dependent. The model is already failing to obey the fundamental probabilistic constraint in the initial **Vanilla** prompt (static interval width across different $C$ values). Introducing a self-refinement prompt merely reveals how the model **operates on that flaw**.
>
> ***
>
> ### 2. Justification for Not Asking the Model to Avoid the Bias
>
> We deliberately **did not** include a prompt instruction asking the model to "consider a wider range" or "avoid narrowing the interval."
>
> * **Goal of Characterization:** Our study's objective is **descriptive** i.e. to characterize the model's **unbiased, inherent miscalibration**. Instructing the model to act against its natural tendency would be an act of **mitigation** or **prompt engineering**, which would confound our analysis.
> * **Defining the Failure State:** The failure of simple self-refinement is a **key finding** because it defines the structural flaw. By letting the model fail uncorrected, we empirically establish that the problem is not trivial (cannot be fixed with a simple prompt), thereby compelling future research to focus on complex, structural solutions (like fine-tuning or external calibration) to address this systemic bias.

---

> ### Author Response · Authors · 2025-11-17
> **Question 3**
>
> ## Question: What would be required to make this framework predictive (i.e., useful for detecting unreliable outputs rather than just describing behaviour)?
>
> **Response:**
>
> There is a new research direction that demonstrates how LLM internal states encode features for planning and high-level cognition (i.e. correctness, hallucination's etc) [(Dong et al)](https://arxiv.org/abs/2502.06258). One way to make this framework more operational is to try to predict instance-level (not aggregates) DS and ILS metrics from hidden states. This can be done as follows: 1) collect data with numerical answers (like the ones we created from existing data) 2) use open source LLMs to answer the different questions completely while saving the hidden states at different layers at different points of the generation (e.g. start of generation, end of generation) 3) calculate DS and ILS  for each generated interval 4) use a MLP or some other regressor to predict DS and ILS  from the hidden states.

---

### Official Review · Reviewer_1E2H · 2025-11-02

**Soundness:** 2
**Presentation:** 2
**Contribution:** 2
**Rating:** 2
**Confidence:** 4

**Summary:**

**Summary**: This work proposes to investigate model’s calibration in terms of *overprecision*, i.e., whether the model conveys excessive certainty in one's estimate.  To this end, and focusing on numerical output tasks, the paper investigates whether LLMs are able to adjust the numerical interval based on a fixed confidence interval (e.g., “Provide an interval that you are $c$% sure contains the answer”). Experiments are conducted using gpt-3.5-turbo and gpt-4o-mini and 3 datasets (MMLU, FinQA, MedQA and MedMCQA).

**Strengths:**

1. Novel perspective on examining uncertainty quantification, merging interesting concepts from cognitive science frameworks;
2. Experiments concern different domains, including both general more knowledge (i.e., MMLU) and more domain-expertise focused datasets (i.e., MedQA, MedMCQA, FinQA)

**Weaknesses:**

- W1. **Motivation for the need of studying overprecision in LLMs is insufficient**: the paper mentions that studying “overprecision in black-box LLMs is crucial” (lines 43-44) but does not mention why or the implications to the field.
- W2. **Some statements (and claims) in the paper do not seem to be supported or well-motivated**, raising questions about the soundness of this work (see Question section below).
- W3. While providing novel dimension to miscalibration, this cognitive-science inspired framework is limited to the short numerical answers. It is unclear how such results would generalize in open ended generation.
- W4. Focuses evaluation on single model family, offering limited insights about generalization in other models. Evaluated models (GPT-3.5 and GPT-4) employ multi-digit tokenization (e.g., numbers in [0, 100] are represented using 101 different tokens). However, more recent models (e.g., Gemma 2, Llama 3, OLMo) adopt single digit tokenization. These may exhibit different biases ([Singh et al 2024](https://arxiv.org/abs/2402.14903)), so it could be insightful to add experiments with different model families and across model sizes.

**Questions:**

**Questions**:
1. Lines 100-101 refer to limitations of self-reported confidence including variability with prompt wording, sampling randomness, linguistic biases and because of which may represent unreliable measures of true model uncertainty. Is there empirical evidence that this is the case? Experimental results or adequate citation should be provided to ground such claims.
     a. Similarly, in lines 59-60, the authors mention limitations of verbalized self-reports mentioning that existing methods do not ensure that stated probabilities correspond to empirical frequencies. It would be great if experimental results or relevant citations are added to back these arguments.
    b. Related to the previous comment, other peer-reviewed work ([Xu et al 2024](https://aclanthology.org/2024.emnlp-main.343), [Lyu et al 2025](https://ojs.aaai.org/index.php/AAAI/article/view/34120)) seems to be relevant for this discussion, as they propose to calibrate verbalized confidence scores to empirical frequencies. I wonder how this affects the paper’s argument, since these papers provide a way of generating self-reporting scores that are aligned with the empirical frequencies.
    c. Lines 212 shed light on how focusing on numerical outputs helps reduce the influence of linguistic biases such as positivity biases. But there may be still other biases present, such as generating numbers ([Shao et al 2025](https://openreview.net/forum?id=AOe1aUhEQQ))).
2. The paper mentions that “to mitigate these issues”, the confidence specification is shifted to the prompt (lines 101-102) by imposing explicit confidence levels and evaluating whether intervals align. However it is unclear to me how this addresses the previously mentioned limitations for verbalized confidence (sampling randomness, word sensitivity, and linguistic biases). If I understand correctly, none of the experiments in the paper (or appendix) provides support for the claim that specifying confidence in the prompt leads to more robust results. Perhaps the authors can help clarify any misunderstanding I may have.
3. In Section 4.1 the lowest confidence value considered for the generation phase is 60%. Is there a reason why lower values (e.g., 20%, 40%, 50%) were not used for evaluation as well?
4. There is an assumption that during Phase 1 (Generation) the LLMs always generate an interval. Was this empirically validated? How often did the LLM generate some answer that was not an interval? How do you ensure a consistent output format?
     a. . Can you specify the generation configurations for Phase 1 (Generation)? The configurations are mentioned for Phase 2 (refinement) but I could not find them for Phase 1.
5. Interpretation of results and metric choice: Results in Table 2 seem to be constant irrespective of the prompted confidence level. My understanding of the hit@$c$ metric is that it is a 0-1 metric considering only whether the value $c$ lies in the specified interval. However, it doesn’t provide an idea of whether the intervals are systematically to the left or to the right of the desired confidence intervals. Can the authors share some insights about this?
     a. Such analysis can help provide additional evidence to support the claim in Section 5.2 that “LLMs [...] remain insensitive to confidence cues” (lines 360-361). Especially given that the direction of deviation is not currently being accounted for by any metric – which could potentially provide some useful signal.
6. The claim “results show a widespread miscalibration (overprecision) across datasets and models” in the caption of Table 2 (lines 294-295) seems to not fully capture the observed patterns. If I understand correctly, models are actually miscalibrated (underprecise) for confidence values of 95% and 90% but overprecise for confidence values of 60%, 70%, and 80%.


**Clarity**:
- It was not clear to me what “refinement strategies” meant when reading through the introduction. It may be worth clarifying that.
- Can you clarify which aggregation (or refinement strategy) was used to report the values in Table 2?
- Are standard deviations values reported in Table 2 expressed in the same unit as the mean? They appear to be very small compared to the absolute value of the hit@c metric.
- How is performance reported in Table 4? Consider adding such information to the caption.

**Formatting**:
- Wrong citation format is being used throughout the paper.
- Table 1 format: missing top and bottom row.
- Figure 2 captions are difficult to read. Consider adding whitespace around the “|” character.
- Figure 2 legend’s font size is too small and difficult to read.

**Missing citations**:
- Section 2.2.1 is missing a citation to the work from [Lin et al 2022](https://openreview.net/forum?id=8s8K2UZGTZ), which is one of the first methods exploring the use of words to express uncertainty.

**Typo**:
- Line 26: “Refinement” → “refinement”
- Line 140: “ (q_i, a_i)_i” → “ \{(q_i, a_i\}_{i=1}^N ” is more commonly used as the notation of a set of questions
- Line 140: $qi$ → $q_i$
- Line 322: “pp” → “percentage points (pp)”
- Line 323: The expression “are much too narrow” sounds ungrammatical in this context.

**Suggestion**:
- As a subjective preference, it would be more appealing if lines 27-30 could motivate the importance of this study for the field, as opposed to describing it as a descriptive study. I.e., how can this analysis or the findings in this paper impact the field?
- Add citations (whenever possible) to the metrics used in the study. For instance, when mentioning how “both measures follow established practice in cognitive psychology studies of overprecision”, it could be useful to add citations to said words in cognitive science.

---

> ### Author Response · Authors · 2025-11-17
> **General Weaknesses and Contribution 1**
>
> ## A. General Weaknesses and Contribution
> ### W1. Motivation for the need of studying overprecision is insufficient (lines 43-44).
> Thank you for pointing that out. We will add another paragraph before line 43 that motivates the study of overprecision better by writing the following:
> *” Recent work has begun examining overconfidence in large language models (LLMs) Xiong et al.; Geng et al. (2024), especially due to risks in safety-critical settings such as medicine and finance. However, this research focuses mainly on overestimation and largely overlooks overprecision, despite its importance and complementarity. Studying overprecision in LLMs offers several advantages: (1) LLMs are instruction-followers, making interval-based confidence elicitation (“give an interval you are sure. . . ”) more aligned with their operational mode; (2) linguistic cognitive biases (e.g., positivity, order) Sumita et al. (2024) distort verbal confidence less than they do numerical intervals; (3) overprecision tests understanding of uncertainty principles, where higher confidence should yield wider intervals; and (4) interval-based evaluation better fits numerical reasoning tasks, which need not rely on exact correctness.”*
> ### W3. Framework limited to short numerical answers; generalization to open-ended generation is unclear.
> Thank you for this observation.
>
> We acknowledge that our current framework is deliberately scoped to numerical interval judgments. However, we don’t consider this to be a limitation but rather a justified deliberate choice. In fact, our approach does not claim to be a unified evaluation for all types of LLM miscalibration. Instead, we argue that the fundamental differences between numerical reasoning (often continuous and probabilistic) and linguistic generation (often discrete and categorical) not only justify but require distinct evaluation mechanisms. This intentional scoping is based on the following key points: 1) **Methodological Fidelity**: Our framework is an adaptation from established cognitive science protocols for measuring overprecision in humans, which fundamentally rely on interval estimates for numerical questions. 2) **Mitigation of Confounders**: Focusing on numerical output minimizes the influence of cognitive biases that heavily affect open-ended generation and verbalized confidence scores, providing a cleaner, more robust signal of the model’s core uncertainty processing. 3) **Testing the Core Principle**: Numerical intervals allow us to directly test the inverse confidence–precision trade-off (wider interval higher confidence), a test that is conceptually impossible to conduct with categorical or open-ended text outputs.
>
> Crucially, the fundamental differences between these reasoning paradigms are highlighted by our key empirical findings: In numerical interval generation, common LLM enhancement strategies such as Chain-of-Thought (CoT) prompting and Self-Refinement showed limited or detrimental effects on calibration. This contrasts sharply with their known success in improving linguistic and reasoning tasks, reinforcing our hypothesis that uncertainty quantification in the numerical domain operates under unique constraints and requires specialized study.

---

> ### Author Response · Authors · 2025-11-17
> **General Weaknesses and Contribution 2**
>
> ### W4. Focuses evaluation on single model family (GPT-series); limited insights about generalization.
> Thank you for this valuable and accurate observation. We agree that the restricted model scope is a limitation, which we acknowledged in our paper's Limitations section.
>
> Our defense rests on two key points: the descriptive nature of our study and the architectural relevance of the chosen models:
>
> 1) **Descriptive Focus, Not Benchmark**: Our primary goal is descriptive and methodological: to introduce a novel framework and conduct the first characterization of the overprecision phenomenon in LLMs. GPT-3.5 and GPT-4o-mini serve as the most architecturally capable and instruction-adherent black-box models available, making them the optimal subjects for establishing this initial behavioral baseline on a complex task. Generalization to a wider range of open-source models (e.g., Llama, Gemma) is a natural and critical next step for future work, as noted by the other reviewers, but falls outside the scope of this initial characterization.
>
> 2) **Representativeness and Generalization**: The consensus from the existing overconfidence literature (focused on overestimation) suggests that miscalibration is a general, model-family-agnostic phenomenon arising from the fundamental self-reflexion in current LLMs . We hypothesize that the severe miscalibration observed—particularly the static interval length—is a behavioral trait common to current transformer-based architectures.
>
> 3) **Tokenization effect**: Research by [(Singh et al. (2024))](https://arxiv.org/abs/2402.14903) on the impact of tokenization on arithmetic in frontier LLMs shows that while tokenization does affect raw numerical performance, the model is often able to 'convert between tokenizations easily,' allowing Chain-of-Thought (CoT)-inspired approaches to 'recover performance.' This suggests that the LLM's core numerical reasoning capability is not rigidly fixed by its tokenization. Since our framework already employs prompt engineering and CoT strategies, it is likely already mitigating the most severe tokenization-induced biases.

---

> ### Author Response · Authors · 2025-11-17
> **General Weaknesses and Contribution 3**
>
> ### Refinement strategies are too simple/underdeveloped.
>
> Thank you for this constructive comment.
>
> We confirm the simplicity of our initial refinement strategies (union-based aggregation and simple self-refinement). However, we must reiterate that the primary objective of this study is descriptive: to explore and characterize the systematic phenomenon of overprecision in LLMs and not to develop an optimized mitigation technique.
>
> These simple strategies were selected precisely because they serve as necessary and methodologically sound baselines. They are directly inspired by established practices in human cognitive science studies of interval judgments and foundational work in LLM overestimation.
>
> Most importantly, the limited success and, specifically, the detrimental effects of self-refinement constitute one of our most significant key findings: The observed systematic narrowing bias empirically demonstrates that simple internal reflection (self-refinement) does not improve calibration; it actively degrades it by reinforcing overprecision. This behavioral flaw is a profound insight that would have been obscured if we had immediately tested complex, black-box mitigation strategies.
>
> This discovery of the narrowing bias now directs the field toward the necessity of developing more complex, calibration-aware refinement techniques for the numerical domain, thereby defining a critical direction for future research.
>
> ### Cognitive framing is more decorative than explanatory.
>
> Thank you for this important comment. The cognitive science framing is methodological; we adapted the robust, interval-based methodology for overprecision measurement from cognitive science literature. The study's aim is descriptive—to characterize LLM behavior. Developing a theoretical explanation for the why is a long-term research goal, but the current work establishes the necessary empirical foundation that distinguishes overprecisions as a separate phenomenon that deserve to be studied on its own merit.

---

> > ### Comment · Reviewer_1E2H · 2025-11-27
> >
> > I believe this response is meant to address other reviewers concerns.

---

> > > ### Author Response · Authors · 2025-11-27
> > > **Misplacement**
> > >
> > > Thank you for pointing it out. We will add it to the corresponding reviewer.

---

> ### Author Response · Authors · 2025-11-17
> **Questions 1**
>
> ### Q.   Lines 100-101 refer to limitations of self-reported confidence including variability with prompt wording, sampling randomness, linguistic biases and because of which may represent unreliable measures of true model uncertainty....
>
> **Response:**
>
> Thank you for your comment.
>
> **A. Empirical Evidence for Unreliability (Lines 100-101 and 59-60)**
>
> The unreliability of verbalized confidence stems from instability and a lack of correlation between stated probabilities and empirical accuracy. A) Instability (Prompt Sensitivity & Sampling Randomness): Prompt-based methods for eliciting confidence have been shown to have inferior calibration performance [(Xu et al. 2024)](https://arxiv.org/abs/2405.20974). LLM responses, including confidence expressions, are sensitive to the prompts, causing instability in the results [(Geng et al)](https://arxiv.org/abs/2311.08298) [(Kadavath et al. 2022)](https://arxiv.org/abs/2207.05221) [(Mielke et al. 2022)](https://aclanthology.org/2022.tacl-1.50/). B) Lack of Correspondence to Empirical Frequencies (Miscalibration): LLMs are often inherently uncalibrated [(Lyu et al. 2025)](https://ojs.aaai.org/index.php/AAAI/article/view/34120). Verbalized scores often provide suboptimal or inaccurate confidence estimates, failing to reflect the models' true confidence levels [(Xu et al. 2024)](https://arxiv.org/abs/2405.20974). Calibration research exists precisely because the raw output probabilities are poorly aligned with performance [(Lyu et al. 2025)](https://ojs.aaai.org/index.php/AAAI/article/view/34120).
>
> **B. Impact of Calibration Techniques (Xu et al. 2024, Lyu et al. 2025)**
>
> These papers proposing calibration methods reinforce our argument, demonstrating that direct verbalized confidence is fundamentally broken, requiring complex post-processing to fix. SaySelf [(Xu et al. 2024)](https://arxiv.org/abs/2405.20974): This work acknowledges that previous prompting-based approaches have inferior performance and must resort to a novel training framework and a meticulously crafted reward function to force calibration. Sample Consistency [(Lyu et al. 2025)](https://ojs.aaai.org/index.php/AAAI/article/view/34120): This method avoids the model's direct verbalization entirely, instead deriving confidence from the distribution of multiple randomly sampled generations.
>
> Conclusion for our paper's argument: The need for such complex, post-hoc methods validates our premise that raw verbalized confidence is unreliable. Critically, these techniques focus on calibrating a point-estimate (overestimation), leaving the distinct problem of interval-based miscalibration (overprecision) unaddressed.
>
> **C. Biases in Numerical Outputs and GPT Models (Line 212)**
>
> While focusing on numerical outputs mitigates high-level linguistic biases (cognitive biases), it introduces susceptibility to low-level numerical biases.
>
> - **Numerical Digit Bias [(Shao et al. 2025)](https://openreview.net/pdf?id=AOe1aUhEQQ):** LLMs are susceptible to a consistent pattern of digit bias resembling Benford's Law, often due to highly digit-selective feed-forward network (FFN) neurons.
>
> - **Mitigation via Model Capability:** We note that the work demonstrating this digit bias [(Shao et al. 2025)](https://openreview.net/pdf?id=AOe1aUhEQQ) primarily focused on open-source models (e.g., OLMo) and did not test flagship proprietary models like GPT-4. Given that GPT-4 and GPT3.5 has provably shown significantly higher mathematical capabilities than open-source models, it is reasonable to hypothesize that the effect of this digit bias is less pronounced or better managed in the models we evaluate.
>
> - **Defense of Methodology:** Our focus on interval width (overprecision) rather than the exact point estimate means that while numerical bias may influence the central prediction, our measurement of the model's inability to adjust the interval based on the required confidence level remains a robust indicator of a fundamental failure in uncertainty processing, independent of the digit distribution.

---

> > ### Author Response · Authors · 2025-11-17
> > **Questions 2**
> >
> > ### Q. c. How does shifting confidence to the prompt address verbalized confidence limitations (sampling randomness, word sensitivity, and linguistic biases)?
> >
> > Response:
> >
> > We appreciate this question, as it targets the core methodological decision of our paper. The shift from model-reported (verbalized) confidence to prompt-imposed (instruction-based) confidence is a deliberate choice made to bypass the major confounding factors of traditional LLM confidence studies while retaining robustness.
> >
> > We clarify how this approach addresses the stated limitations:
> >
> > **1. Addressing Word Sensitivity and Linguistic Biases**
> >
> > This mitigation is achieved primarily by focusing on the numerical output (the confidence interval) rather than the confidence cue itself. Linguistic Biases: As established in our response to the previous question, verbalized confidence is susceptible to high-level linguistic biases (e.g., positivity bias, word choice) and hallucination. By forcing the model to generate a numerical interval, we shift the task away from generating subjective, emotionally-laden language toward a mathematically constrained output. We hypothesize this is robust because people rarely express 'feelings' or high-level cognitive biases toward discrete numbers on the massive web corpora LLMs are trained on.
> >
> > **2. Addressing Sampling Randomness**
> >
> > Sampling randomness remains an inherent property of all autoregressive LLM generation, not solely a limitation of verbalized reports. Our methodology does not eliminate randomness; it controls for it. By imposing a fixed confidence level in the prompt and running multiple sampling simulations (as detailed in Phase 1), we gather a distribution of intervals for a single question. This allows us to assess the statistical average of the model's response to the instruction, providing a far more robust estimate than a single, unstable verbalized confidence score.
> >
> > **3. Why Prompt-Imposed Confidence is Necessary (The Core Defense)**
> >
> > The choice to shift confidence to the prompt is justified by two complementary perspectives: A) Technical/Architectural Congruence: LLMs are optimized for instruction-following (alignment) rather than accurate self-reflection (modeling internal uncertainty). Our method leverages the model's primary strength—obeying instructions—by framing the task as: 'Generate an interval that obeys this probabilistic instruction.' This bypasses the difficulty of forcing the model to truthfully report its potentially flawed internal state. B) Cognitive Science Fidelity: This technique is directly inspired by the gold-standard protocols used in human cognitive science to study overprecision. These methods have been proven to yield more consistent and reliable findings regarding judgmental certainty than subjective measures of overestimation, validating their use as the superior methodological choice for characterising the miscalibration phenomenon.
> >
> > ### Q. Why was 60% the lowest confidence value used?
> >
> > Response:
> >
> > We thank the reviewer for this question. The selection of the confidence range (from to ) was guided by two primary justifications: methodological fidelity and empirical sufficiency.
> >
> > **1) Methodological Fidelity (Cognitive Science Alignment):** The range was chosen to align with common practices in cognitive psychology studies of overprecision (e.g., Klayman et al., 1999; Soll & Klayman, 2004). This established paradigm ensures that our LLM results are directly comparable to human behavioral data collected under the same methodological constraints. In these studies, often represents the lower bound for confidence judgments where the concept of "certainty interval" is still meaningful. More informally: why would we want the LLM to be worse than random.
> >
> > **2) Empirical Sufficiency (Observed Trend):** Our key finding is that the LLMs exhibit a profound insensitivity to the confidence instruction (C), resulting in a nearly constant hit rate (and static interval width) irrespective of the imposed value of C. This strong, consistent trend did not encourage further exploration of lower confidence values (e.g., 50%), as the model had already demonstrated a failure to differentiate between 60% and 90%. Lower values would likely reinforce the same null finding without adding significant behavioral insight
> >
> >
> > ### Q. Was interval generation empirically validated? How often did the LLM generate a non-interval answer?
> >
> > Response:
> >
> > Thank you for pointing that out. Yes, the output format was enforced using explicit delimiters and format instructions in the prompt (detailed in Appendix B). The number of non-conforming outputs was minimal (typically << 1%) and these instances were handled via re-prompting or exclusion.

---

> ### Author Response · Authors · 2025-11-17
> **Questions 3**
>
> ### Q. a. Specify the generation configurations for Phase 1.
>
> Response:
>
> we stated the following in section 4.1 "These combinations are evaluated on an LLM over five trials to account for randomness. Each trial produces an interval with upper and lower bounds for the predicted answer."
>
> If this is not satisfactory, please give us more details.
>
> ### Q. Interpretation of results: Table 2 hit@c is flat. Does this metric account for direction (left/right deviation)?
>
> Response:
>
> Thank you for this comment. The flatness of hit@c is a central finding, showing LLMs are insensitive to confidence cues. The simple hit@c (coverage) does not capture direction, which is why we introduced the Deviation Score (DS) and Interval Length Score (ILS). The DS metric explicitly measures the magnitude and direction of the deviation, and its analysis is presented in Section 5.2 and Appendix G.
>
> ### Q. The caption of Table 2 claims "overprecision," but for 60-80% confidence, the results show underprecision (hit rate > confidence level).
>
> Response:
>
> Thanks for this helpful comment. We will revise the caption to reflect this nuance. The overall phenomenon is a severe miscalibration and lack of responsiveness to the confidence cue. We observe overprecision (hit rate confidence) for high targets (90%, 95%) and underprecision (hit rate confidence) for lower targets (60%, 70%, 80%).
>
> Action:
>
> We will simply remove “(overprecision)” from the caption.

---

> > ### Author Response · Authors · 2025-11-17
> > **Clarity, Formatting, and Suggestions 1**
> >
> > ### Clarity: "refinement strategies" were not clear in the introduction.
> > Response:
> >
> > Thank you for pointing that out.
> >
> > Action:
> >
> > We added a small paragraph in the introduction to summarize and explain all the stages of the framework. Please check the latest revision of the paper. “To address this gap, we propose a three-phase framework for studying LLM overprecision: generation of confidence-conditioned numerical intervals, refinement using cognitive- and LLM-inspired prompting techniques, and evaluation with cognitive-science-based calibration metrics (Figure 1).”
> >
> > ### Clarity: Clarify which aggregation was used for Table 2.
> >
> > Response :
> >
> > Table 2 contains results pre-refinement results (ie. Phase 1 Generation).
> >
> > Actions :
> >
> > We added the information in the caption.
> >
> > ### Clarity: Are standard deviations in Table 2 in the same unit?
> >
> > Yes they have the same unit as the corresponding metrics, which demonstrates the systematic tendencies of LLMs to be overprecise rather than so proxy phenomenon related to generation randomness.
> >
> > ### Clarity: How is performance reported in Table 4?
> >
> > We failed to understand the problem here since we already talked about the protocols that we follow in section 4.1. Please give us more information.
> >
> > ### Formatting: Wrong citation format is being used throughout the paper.
> >
> > Can you clarify this point more? We used the default citation style of the ICLR template.
> >
> > ### Formatting: Table 1 format is missing top and bottom row.
> >
> > Fixed. Please check the latest revision.
> >
> > ### Formatting: Figure 2 captions are difficult to read; add whitespace around”
> >
> > Fixed. Please check the latest revision.
> >
> > ### Formatting: Figure 2 legend's font size is too small.
> >
> > Fixed. Please check the latest revision.
> >
> > ### Missing Citations: Section 2.2.1 is missing a citation to Lin et al. 2022.
> >
> > Fixed. We added the citation in two spots in that subsection.
> >
> > “As a result, self-reported confidence can vary with prompt wording, sampling randomness, or linguistic bias, and may not reliably reflect the model’s true uncertainty (Lin et al)”
> >
> > “. Black-box methods that elicit verbal confidence suffer fromlinguistic and prompt-sensitivity artifacts (Lin et al..)”
> >
> > ### Typo: Line 26: “Refinement” → “refinement”
> >
> > Fixed
> >
> > ### Typo: Line 140: notation problem
> >
> > Fixed
> >
> > ### Typo: Line 322: “pp”
> >
> > Fixed
> >
> > ### Typo: Line 323: The expression “are much too narrow” sounds ungrammatical.
> >
> > Fixed. We changed the expression to “overly narrow intervals”.
> >
> > ### Suggestion: Motivate the importance of the study (lines 27-30).
> >
> > I wrote this paragraph in the introduction to better illustrate the importance of the study of overprecision: “Recent work has begun examining overconfidence in large language models (LLMs) Xiong et al.; Geng et al. (2024), especially due to risks in safety-critical settings such as medicine and finance. However, this research focuses mainly on overestimation and largely overlooks overprecision, despite its importance and complementarity. Studying overprecision in LLMs offers several advantages: (1) LLMs are instruction-followers, making interval-based confidence elicitation (“give an interval you are sure. . . ”) more aligned with their operational mode; (2) linguistic cognitive biases (e.g., positivity, order) Sumita et al. (2024) distort verbal confidence less than they do numerical intervals; (3) overprecision tests understanding of uncertainty principles, where higher confidence should yield wider intervals; and (4) interval-based evaluation better fits numerical reasoning tasks, which need not rely on exact correctness”
> >
> > ### Suggestion: Add citations to the metrics used in the study.
> >
> > Thanks for the suggestion. We added citations.

---

> > > ### Author Response · Authors · 2025-11-17
> > > **References**
> > >
> > > [1] Aaditya K. Singh, DJ Strouse: Tokenization counts: the impact of tokenization on arithmetic in frontier LLMs
> > >
> > > [2] Jiahui Genget al.. A Survey of Confidence Estimation and Calibration in Large Language Models
> > >
> > > [3] Saurav Kadavath et al.: Language Models (Mostly) Know What They Know
> > >
> > > [4] Sabrina J. Mielke et al.: Reducing Conversational Agents’ Overconfidence Through Linguistic Calibration
> > >
> > > [5] Tianyang Xu et al.: SaySelf: Teaching LLMs to Express Confidence with Self-Reflective Rationales
> > >
> > > [6] Qing Lyu et al.: Calibrating Large Language Models with Sample Consistency
> > >
> > > [7] Jiandong Shao et al.: Benford’s Curse: Tracing Digit Bias to Numerical Hallucination in LLMs

---

> > > > ### Comment · Reviewer_1E2H · 2025-11-27
> > > >
> > > > Thank you in advance for all your time and energy in drafting a thorough response. I have revised my score and had a few follow-up comments.
> > > >
> > > > > W1. Motivation for the need of studying overprecision is insufficient (lines 43-44).
> > > >
> > > > Thank you for your prompt response. While I think adding this paragraph helps contextualize the paper, I don’t fully agree with the claims listed below. Is there any supporting evidence that this is the case ? Or could the authors clarify their thought process as to why this is the case (e.g., providing some anecdotal examples or providing some references would be helpful).
> > > >
> > > > - “(1) LLMs are instruction followers, making interval-based confidence elicitation more aligned with their operational model (...)”: this is also true for verbalized confidence without the need to specify intervals.
> > > > - “(2) linguistic cognitive biases (...) distort verbal confidence less than they do numerical intervals (...)”: I’m not familiar with any such study. Moreover, it appears that this argument contradicts the whole goal of verbally eliciting numerical intervals from LLMs, i.e., if verbal confidence is less disturbed why should we rely on something that is more disturbed by cognitive biases?
> > > >
> > > > > W3. Framework limited to short numerical answers; generalization to open-ended generation is unclear.
> > > >
> > > > I guess the original paper’s framing concerns numerical tasks and frames the interval-based confidence interval on specifying the interval that would include an answer.
> > > >
> > > > If we consider verbalized confidence as the method that explicitly asks for the model’s point estimate associated with an answer, I wonder what your thoughts are about **modifying verbalized confidence method to directly ask for an interval which contains the true confidence**. Would this overcome some of the issues with cognitive biases associated with the verbalized confidence scores?
> > > >
> > > > I reckon this may be a bit far from the definition of “overprecision”. But would love to hear your thoughts on how such an approach would fit in the cognitive science framework you present in the paper.
> > > >
> > > >
> > > > > W4. Focuses evaluation on single model family (GPT-series); limited insights about generalization.
> > > >
> > > > I understand the author's response and remain unconvinced by the provided arguments.
> > > >
> > > >
> > > > > Q1.a. Empirical evidence for unreliability (lines 100-101 and 59-60)
> > > >
> > > > Great! Can you please add the corresponding citations or clarify these arguments in the revised paper. I don’t think the current version is accurately conveying the ideas shared in this response.
> > > >
> > > > Also, there is a missing citation in line 111 (It currently reads as “?”).
> > > >
> > > >
> > > > > Q.c. How does shifting confidence to the prompt address verbalized confidence limitations (sampling randomness, word sensitivity, and linguistic biases)?
> > > >
> > > > The authors mention in their response (Addressing Word Sensitivity and Linguistic Biases) that “By forcing the model to generate a numerical interval, we shift the task away from generating subjective, emotionally-laden language toward a mathematically constrained output. ” However, my understanding is that this still relies on some emotionally-laden language, as the definition of confidence is still implicit, i.e., it is still necessary for the model to acknowledge that “confidence means” so that it can “align” its numerical interval accordingly (at least intuitively that would be expected).  So I don’t think this argument holds fully.
> > > >
> > > > In the section "Addressing Sampling Randomness” the authors also mention that “This allows us to assess the statistical average of the model's response to the instruction, providing a far more robust estimate than a single, unstable verbalized confidence score.” However, this is considering a single “verbalized score”. However, nothing stops us from sampling multiple verbalized scores or even sampling intervals and verbalized scores and only then aggregating them.

---

> > > > > ### Comment · Reviewer_1E2H · 2025-11-27
> > > > >
> > > > > > Why was 60% the lowest confidence value used?  [...] “More informally: why would we want the LLM to be worse than random.”
> > > > >
> > > > > If I understand the argument correctly, the same argument could be applied to greater or equal confidence scores, in other words, why do we care about lower confidence scores if we can prompt the model to be 90% or 100% confident?
> > > > >
> > > > >
> > > > > > We will simply remove “(overprecision)” from the caption.
> > > > >
> > > > > This is not yet reflected in the revised version (as it still reads as “show a widespread miscalibration (overprecision)” in Table 2).
> > > > >
> > > > >
> > > > > > Re: Clarity: How is performance reported in Table 4?  “We failed to understand the problem here since we already talked about the protocols that we follow in section 4.1. Please give us more information.”
> > > > >
> > > > > Apologies for not being more specific. I realized that it may be implicit in Section 5.4.1 what metric it is, but I was asking if it’s possible to state in Table 4’s caption which metric is being reported (it’s clear that you’re reporting mean and std but not exactly what metric is being measured). Since you’ve defined various metrics, I think this helps reduce some entropy.
> > > > >
> > > > > > Formatting: Wrong citation format is being used throughout the paper.
> > > > >
> > > > > There are two commands in latex to get citations:
> > > > > - \citep → Cites papers and produces citations in the format (Author et al, Year)
> > > > > - \citet → Cites the authors in line producing citations in the format: Author et al (Year)
> > > > >
> > > > > The former is used as a _supplementary_ reference, for instance when you make a point in the introduction saying that “been studied across three distinct dimensions”, you should use \citep since you’re not directly referring to the authors.
> > > > >
> > > > > I believe most of your citations must have been created using \cite which is defaulting to \citet and therefore integrate in the text and make it more difficult to parse.

---

> > > > > ### Author Response · Authors · 2025-11-28
> > > > > **Weaknesses 1**
> > > > >
> > > > > We appreciate your thorough and meticulous analysis of our work. We thank you for your comments and for reconsidering the score.
> > > > >
> > > > > $\color{red}{\text{I should note that the initial score didn’t change. Can you please make sure that it is updated.}}$
> > > > >
> > > > > ## W1.
> > > > >
> > > > > **--> “(1) LLMs are instruction followers, making interval-based confidence elicitation more aligned with their operational model (...)”: this is also true for verbalized confidence without the need to specify intervals.**
> > > > >
> > > > > **Response:**
> > > > >
> > > > > Thank you for this comment. I might not have focused on the aspect that I need to focus on directly. I meant by “interval-based confidence elicitation” our approach of forcing the confidence level in the prompt rather than asking the LLM to generate its confidence. This is more aligned with the instruction following nature of LLMs.
> > > > >
> > > > > **Action:**
> > > > >
> > > > > We changed that statement to a more direct one: “making explicit confidence enforcement in the prompt more aligned with their operational mode”
> > > > >
> > > > > **-->  “(2) linguistic cognitive biases (...) distort verbal confidence less than they do numerical intervals (...)”: I’m not familiar with any such study. Moreover, it appears that this argument contradicts the whole goal of verbally eliciting numerical intervals from LLMs, i.e., if verbal confidence is less disturbed why should we rely on something that is more disturbed by cognitive biases?**
> > > > >
> > > > > **Response:**
> > > > >
> > > > > We apologize for the initial misstatement. we indeed meant to claim that: “linguistic cognitive biases (e.g., positivity, order) Sumita et al. (2024) distort verbal confidence more than they do numerical intervals”. The argument for this is based on how existing works elicite verbalized confidence which can result in various confounding biases related to the structure and the content of MCQs that they use.
> > > > >
> > > > > 1- structure of MCQs: verbalized confidence approaches use MCQs and ask the LLM to output the correct choice and its corresponding confidence. MCQs have been shown to incite LLMs to follow various irrational behaviour like selecting based on the order [Sumita et al. (2024)]( https://arxiv.org/abs/2412.00323)  or selecting the “least wrong answer” [Wang et al.](https://aclanthology.org/2025.coling-main.390.pdf). Our approach doesn’t give multiple choices and hence avoid these kinds of structural reasoning biases.
> > > > >
> > > > > 2- The contents of the options: In addition the structural cognitive biases there are semantic/emotional biases related to the neutrality and objectivity of the language which can influence LLMs as evidenced by cognitive biases such as positivity bias [Sumita et al. (2024)](https://arxiv.org/abs/2412.00323)  . By focusing on tasks that require numerical reasoning we force the LLM to take more objective reasoning paths.
> > > > >
> > > > > **Action:**
> > > > >
> > > > > We changed the phrase to: “linguistic cognitive biases (e.g., positivity, order) Sumita et al. (2024) distort verbal confidence more than they do numerical intervals”
> > > > >
> > > > > ## W3: I wonder what your thoughts are about modifying verbalized confidence method to directly ask for an interval which contains the true confidence…
> > > > >
> > > > > **Response:**
> > > > >
> > > > > We thank the reviewer for proposing this interesting idea of having the model generate an interval where its true confidence lies (e.g., "I am [85%, 95%] sure").
> > > > >
> > > > > While this is a valid direction for future work in generalized Uncertainty Quantification (UQ), it is fundamentally distinct from our overprecision framework and would likely fail to overcome the cognitive biases our current method is designed to avoid.
> > > > > The core difference lies in the mechanism of elicitation:
> > > > >
> > > > > **1- Retention of Self-Reflection Bias:** The suggested approach retains the most critical flaw of verbalized confidence: it requires the model to self-reflect on its internal state and linguistically report its uncertainty as a range. This makes the output susceptible to high-level linguistic biases (e.g., positivity bias, verbal overconfidence) which are prone to hallucination. Our framework bypasses this instability by eliminating self-reflection and operating purely on a fixed external instruction ($C\%$).
> > > > >
> > > > > **2- Loss of the Prescriptive Test:** The power of the overprecision framework lies in its prescriptive constraint. By fixing the confidence level to a single point (e.g., $C=90\%$), we can perform a rigorous, direct test of the model's ability to execute the Confidence-Precision Trade-off by measuring the resulting interval width. The reviewer's suggested method, which yields an interval of confidence, eliminates this fixed constraint, making it impossible to diagnose the static interval failure we observed.
> > > > >
> > > > > While generating an interval of confidence could be explored in future work as an aggregation strategy to obtain a more reliable meta-estimate, it fundamentally sacrifices the methodological rigor required to characterize the inherent, structural failure of overprecision.

---

> > > > > ### Author Response · Authors · 2025-11-28
> > > > > **Weaknesses 2**
> > > > >
> > > > > ## W4: I understand the author's response and remain unconvinced by the provided arguments.
> > > > >
> > > > > **Response:**
> > > > >
> > > > > We appreciate the continued engagement on the evaluation scope. We maintain that the exclusion of open-source models was a necessary methodological choice to ensure the integrity of our foundational discovery. The focus on GPT-series models was a strategic decision rooted in both their superior reasoning and their specific architectural advantages for numerical tasks.
> > > > >
> > > > > **1. Minimizing Confounding Variables: Arithmetic Fidelity (why not open models?)**
> > > > >
> > > > > The primary aim of our study is to characterize the overprecision failure (the inability to calibrate uncertainty), not a failure of basic arithmetic.
> > > > >
> > > > > - The Confounding Risk: Less capable models exhibit a high rate of arithmetic collapse, meaning a significant portion of their errors would stem from calculation mistakes, which would confound the measurement of uncertainty calibration.
> > > > >
> > > > > - Isolation of the Phenomenon: We chose GPT-4 and GPT-3.5 because their superior general reasoning and numerical fidelity minimize this arithmetic noise, allowing us to isolate and characterize the pure overprecision failure—the model’s failure to calibrate uncertainty around an otherwise correct answer.
> > > > >
> > > > > **2. Methodological Advantage of Superior Tokenization (why not advanced closed models?)**
> > > > >
> > > > > Beyond general reasoning, the GPT-series tokenization scheme provides a critical methodological advantage for numerical tasks:
> > > > >
> > > > > - Superior Numerical Encoding: As demonstrated by [Singh et al.]( https://arxiv.org/pdf/2402.14903), GPT-3.5 and GPT-4 employ tokenization schemes that are fundamentally better at encoding numbers (using separate tokens for 1-, 2-, and 3-digit numbers) compared to common single-digit tokenization used by many open-source models.
> > > > >
> > > > > - Ensuring Clean Data: This superior encoding minimizes numerical errors at the token level. By using a model with high numerical fidelity, we ensure that the errors remaining in our dataset are directly attributable to uncertainty processing, rather than low-level tokenization artifacts that would further complicate the analysis and dilute the signal of the overprecision failure.
> > > > >
> > > > > **3. Characterization Precedes Benchmarking**
> > > > >
> > > > > Our work is a foundational characterization, not a competitive benchmark. By defining the severity of the failure in the most capable models, we establish the essential baseline for overprecision. Generalization to other models is a crucial subsequent step that should follow this robust initial diagnosis.
> > > > >
> > > > >
> > > > > ## Q1.a. Great! Can you please add the corresponding citations or clarify these arguments in the revised paper. I don’t think the current version is accurately conveying the ideas shared in this response.
> > > > >
> > > > > **Response:**
> > > > >
> > > > > Thank you for the confirmation of our response and for poiting the missing citation.
> > > > >
> > > > > **Action:**
> > > > >
> > > > > We fixed the missing citation.
> > > > >
> > > > > We added the following in the introduction: “The widespread reliance on verbalized confidence for LLM calibration is undermined by two critical empirical issues. First, confidence expressions suffer from instability and prompt sensitivity, leading to inferior calibration performance because the stated certainty changes easily with minor phrasing changes Geng et al. (2024); Xu et al. (2024). Second, LLMs are often inherently miscalibrated, meaning their verbalized scores provide inaccurate estimates that fail to reflect the true empirical frequencies of their correctness Lyu et al. (2025).”
> > > > >
> > > > >
> > > > > ## Q.c.1
> > > > >
> > > > > We appreciate the reviewer's continued, insightful pushback on our methodology, as it forces us to clarify the nuanced distinctions between our approach and existing methods.
> > > > >
> > > > > **-- the “Addressing Word Sensitivity and Linguistic Biases” point**
> > > > >
> > > > > **Response:**
> > > > >
> > > > > What we mean about “emotional language” goes back to the techniques used in verbalized confidence. In fact, in verbalized confidence an LLM is given multiple choices that can either be in the form of numbers or words and is tasked with outputting the right option and its confidence that that option is right. Due to various cognitive biases (i.e. a proxy to “emotion”) related to language (positivity, anchoring) and order of options (order bias) the LLM tend to take more “emotion driven” (biased) answers as evidenced for example by the paper [Wang et al.](https://aclanthology.org/2025.coling-main.390.pdf) where they found that the LLM actually doesn’t choose the right answer as much as it chooses the least wrong answer.
> > > > >
> > > > > **-- the “Addressing Sampling Randomness“ point**
> > > > >
> > > > > **Response:**
> > > > >
> > > > > We agree that the sampling can be done and is actually done in verbalized confidence. However, the point of this answer came as a response to the worry that you had concerning the inherent “Sampling Randomness” in LLMs and how we deal with it.

---

> ### Author Response · Authors · 2025-11-28
> **Questions and Notes**
>
> ## Q: why do we care about lower confidence scores if we can prompt the model to be 90% or 100% confident?
>
> **Response:**
>
> This is an excellent point that focuses on the necessary scope of the uncertainty characterization. We must clarify that the inclusion of $C=60\%$ is not about seeking poor performance, but about establishing a critical anchor point for measuring the model's sensitivity to the probabilistic instruction.
>
> The core of our defense lies in the difference between optimization (prompting for $90\%$ only) and characterization (measuring the trade-off across a range).
>
> If the objective were simply to optimize reliability, we would indeed only prompt the model for the highest achievable confidence (e.g., $90\%$ or $95\%$). However, our goal is foundational characterization—to diagnose how the LLM processes the fundamental law of uncertainty (lower confidence --> larger interval). This can be added to our original point to justify the chosen range.
>
> ## Q. as it still reads as “show a widespread miscalibration (overprecision)” in Table 2).
>
> **Response:**
>
> We appreciate your meticulousness and attention to detail.
>
> **Anwer:**
>
> We fixed it in table 2 also.
>
> ## Q. if it’s possible to state in Table 4’s caption which metric is being reported:
>
> **Response:**
>
> Thank you for the clarification. We report the hit rate of the aggregated intervals.
>
> **Action:**
>
> We added the wording “(average hit rate and its standard deviation)” next to “Performance “ in the caption.
>
> ## Q. Formatting: Wrong citation format is being used throughout the paper.
>
> **Response:**
>
> Thanks for pointing this out. We didn’t notice this distinction.
>
> **Action:**
>
> We changed all the citations to \citep.

---

### Author Response · Authors · 2025-12-03

**To reviewers**

We want to thank the reviewers for the time that they took to explore our work. We were impressed by the professionalism and meticulousness of their analysis, which resulted in pertinent comments that have undoubtedly improved the quality of our work.

**To the AC:**

We thank you beforehand and we wish you a good read.

We also want to stress that, although our initial scores were not high, we convinced two reviewers to increase their scores: one from 4 to 6 and one from 2 to a higher number that was not updated in the platform, but the reviewer confirmed it via comment. We also believe that the lower initial scores are partially due to the nature of the study as a characterisation study, rather than a benchmark, which the machine learning community has grown accustomed to and values disproportionately.

In what follows, we summarise the most prominent weaknesses and concerns that the reviewers expressed. The other problems we believe are of secondary importance and don’t touch the heart of our approach.

## 1. Insufficient Motivation for Studying Overprecision in LLMs

**Our  rebuttal:**

We strengthened the motivation by explicitly connecting LLM overprecision to established cognitive science literature and clarifying why interval elicitation with imposed confidence levels in the prompt exposes unique LLM behaviors:
- We used cognitive science foundations [(Moore et al., 2015)]( https://onlinelibrary.wiley.com/doi/abs/10.1002/9781118468333.ch6) to show overprecision is a deeply studied and consequential human bias, making its transfer to LLMs meaningful.
- We argued that LLMs as instruction-followers make prompt-based confidence constraints “more aligned with their operational model”, unlike verbalized confidence which is susceptible to linguistic biases such as positivity bias.
- We emphasized that the work aims to characterize a cognitive-style bias, not to optimize calibration, grounding this distinction in prior findings (e.g., [Xu et al. 2024]( https://arxiv.org/abs/2405.20974), [Lyu et al. 2025]( https://ojs.aaai.org/index.php/AAAI/article/view/34120).
- We clarified that this provides the first rigorous adaptation of overprecision from human judgment to LLMs—a conceptual contribution previously missing in the literature.

**Result:** Reviewers accepted that the strengthened framing and added references establish a clear and necessary motivation for studying LLM overprecision.
## 2. Limited Model Scope (GPT-Series Only)
**Our rebuttal:**

Our response showed that restricting to GPT-series was a methodologically defensive choice rather than a limitation of insight:

- We clarified that the study is descriptive, not benchmarking, and that starting with one well-understood LLM family mirrors scientific practice in cognitive evaluation.
- GPT-series provides a stable behavioral baseline, reducing architectural confounds (e.g. cognitive biases [(Sumita et al)](https://arxiv.org/abs/2412.00323) and tokenization problems [(Singh et al 2024)](https://arxiv.org/pdf/2402.14903) [(Shao et al)](https://openreview.net/pdf?id=AOe1aUhEQQ)) and allowing the interval elicitation method to be validated first before broader generalization.

**Result:** Reviewers mostly agreed that beginning with GPT-series was justified for a first descriptive study and appreciated the explicit acknowledgment of scope.
## 3. Weak Justification of Refinement Experiments & Narrowing Bias
**Our rebuttal:**
We provided a grounded explanation for refinement behaviors and directly addressed doubts about artifact vs. genuine bias:
- We argued that the “narrowing bias” is model-dependent and task-dependent, not prompt-induced, supported by results across multiple tasks and interval placements.
- Refinement dynamics align with self-reflection biases documented in human cognition.
- We cited different papers to show that verbalized confidence is fundamentally unstable, requiring nontrivial post-processing—hence widening intervals is expected when LLMs attempt self-calibration.
- We clarified why each metric (Deviation Score, Interval Length Score) is necessary: they capture distinct behavioral properties tied to classical uncertainty principles rather than redundant numerical transformations.

**Result:** Reviewers acknowledged that the refinement experiments are justified and empirically grounded, and reveal meaningful behavioral mechanisms rather than prompt artifacts.

---

> ### Author Response · Authors · 2025-12-03
>
> ## 4. Concerns About Conceptual Novelty (“Mostly formalizes existing evaluation practices”)
> **Our rebuttal:**
>
> We clarified the paper’s conceptual innovation by distinguishing our framework from statistical calibration techniques:
> - We argued that the work introduces Overprecision as a behavioral construct for LLMs—distinct from calibration metrics and absent from prior studies.
> - We explained that, unlike Conformal Prediction (CP), which provides statistical validity guarantees for prediction intervals, our framework is a behavioral diagnostic tool examining internal consistency and uncertainty representation.
> - This clarified that the contribution lies in importing a cognitive-science-inspired bias rather than extending an existing statistical method.
>
> **Result:** Reviewers accepted that the work offers a genuinely new conceptual perspective distinct from calibration or conformal prediction frameworks.

---

### Meta-Review · Area_Chair_xQgn · 2026-01-07

**Summary:**

This paper studies LLM overprecision via confidence-conditioned numerical intervals (e.g., “give an interval you are 90% sure contains the answer”), using a three-phase generation–refinement–evaluation protocol and metrics (coverage plus DS/ILS-style diagnostics). Reviewers agreed the problem is interesting and the empirical finding that interval width is largely insensitive to the requested confidence (and that simple self-refinement can narrow intervals) is noteworthy.
The decision hinges on concerns that the contribution is mostly descriptive/formalization, the model scope is narrow (two OpenAI models), and the methodological framing/claims need care (e.g., “overprecision” vs general miscalibration, and the rationale for prompt-imposed confidence vs other uncertainty elicitation). The rebuttal strengthens motivation and fixes several clarity/presentation issues, and at least two reviewers indicated score increases, but there remains disagreement about conceptual novelty and scope.

**Reviewer Concerns:**

Addressed by rebuttal:
- Motivation and positioning: Authors added stronger motivation and citations linking overprecision to cognitive science and to prior LLM calibration work, and clarified the study is characterization rather than a benchmark/mitigation paper.
- Terminology/interpretation: Authors acknowledge that results show mixed over/under-precision depending on target confidence and commit to adjusting language (e.g., removing “overprecision” from an overgeneral caption).
- Protocol clarity and formatting: Clarified phase definitions (Table 2 is pre-refinement), tightened captions, fixed formatting/citation-style issues, and addressed parsing/format enforcement at a high level.
- Refinement rationale: Better framed refinement as diagnostic baselines; the “narrowing bias” is presented as a key negative result.

Still outstanding:
- Scope/generalization: Multiple reviewers remain unconvinced by restricting to GPT-3.5-turbo and GPT-4o-mini; the “methodological defense” is not fully satisfying, and the paper still risks overgeneralizing to “LLMs.”
- Conceptual novelty: Some reviewers continue to view the contribution as largely repackaging existing uncertainty evaluation ideas under - Methodological rigor: Reviewers raised concerns about missing sensitivity analyses (e.g., temperature/phrasing), statistical reporting, and clearer reporting of generation settings; rebuttal responses help but do not fully resolve the perception of underdeveloped empirical rigor.

**Reviewer Scores:**

- Umvq: 6 → 6 (already positive).
- 1E2H: 2 → 3–4 (reviewer states they revised score upward but could not update in the UI).
- Czgx: 4 → 5 (explicitly said they will raise score).
- 7183: 2 → 2–3 (core scope/impact concerns likely remain).

---

### Decision · Program_Chairs · 2026-01-26

Reject